# Online Influence Maximization under Independent Cascade Model with Semi-Bandit Feedback

**Zheng Wen**
Adobe Research
zwen@adobe.com

**Branislav Kveton**
Adobe Research
kveton@adobe.com

**Michal Valko**
SequeL team, INRIA Lille - Nord Europe
michal.valko@inria.fr

**Sharan Vaswani**
University of British Columbia
sharanv@cs.ubc.ca

## Abstract

We study the online influence maximization problem in social networks under the *independent cascade* model. Specifically, we aim to learn the set of "best influencers" in a social network online while repeatedly interacting with it. We address the challenges of (i) combinatorial action space, since the number of feasible influencer sets grows exponentially with the maximum number of influencers, and (ii) limited feedback, since only the influenced portion of the network is observed. Under a stochastic semi-bandit feedback, we propose and analyze `IMLinUCB`, a computationally efficient UCB-based algorithm. Our bounds on the cumulative regret are polynomial in all quantities of interest, achieve near-optimal dependence on the number of interactions and reflect the *topology* of the network and the *activation probabilities* of its edges, thereby giving insights on the problem complexity. To the best of our knowledge, these are the first such results. Our experiments show that in several representative graph topologies, the regret of `IMLinUCB` scales as suggested by our upper bounds. `IMLinUCB` permits linear generalization and thus is both statistically and computationally suitable for large-scale problems. Our experiments also show that `IMLinUCB` with linear generalization can lead to low regret in real-world online influence maximization.

## 1 Introduction

Social networks are increasingly important as media for spreading information, ideas, and influence. Computational advertising studies models of information propagation or diffusion in such networks [16, 6, 10]. *Viral marketing* aims to use this information propagation to spread awareness about a specific product. More precisely, agents (marketers) aim to select a fixed number of influencers (called *seeds* or *source nodes*) and provide them with free products or discounts. They expect that these users will influence their neighbours and, transitively, other users in the social network to adopt the product. This will thus result in information propagating across the network as more users adopt or become aware of the product. The marketer has a budget on the number of free products and must choose seeds in order to maximize the *influence spread*, which is the expected number of users that become aware of the product. This problem is referred to as *influence maximization* (IM) [16].

For IM, the social network is modeled as a directed graph with the nodes representing users, and the edges representing relations (e.g., friendships on Facebook, following on Twitter) between them. Each directed edge $(i, j)$ is associated with an *activation probability* $\overline{w}(i, j)$ that models the strength of influence that user $i$ has on user $j$. We say a node $j$ is a *downstream neighbor* of node $i$ if there is a directed edge $(i, j)$ from $i$ to $j$. The IM problem has been studied under a number of

diffusion models [16, 13, 23]. The best known and studied are the models in [16], and in particular the *independent cascade* (IC) model. In this work, we assume that the diffusion follows the IC model and describe it next.

After the agent chooses a set of source nodes $\mathcal{S}$, the independent cascade model defines a diffusion (influence) process: At the beginning, all nodes in $\mathcal{S}$ are activated (influenced); subsequently, every activated node $i$ can activate its downstream neighbor $j$ with probability $\overline{w}(i, j)$ once, *independently* of the history of the process. This process runs until no activations are possible. In the IM problem, the goal of the agent is to *maximize the expected number of the influenced nodes* subject to a cardinality constraint on $\mathcal{S}$. Finding the best set $\mathcal{S}$ is an NP-hard problem, but under common diffusion models including IC, it can be efficiently approximated to within a factor of $1 - 1/e$ [16].

In many social networks, however, the activation probabilities are *unknown*. One possibility is to learn these from past propagation data [25, 14, 24]. However in practice, such data are hard to obtain and the large number of parameters makes this learning challenging. This motivates the learning framework of IM bandits [31, 28, 29], where the agent needs to learn to choose a good set of source nodes *while* repeatedly interacting with the network. Depending on the feedback to the agent, the IM bandits can have (1) full-bandit feedback, where only the *number of influenced nodes* is observed; (2) node semi-bandit feedback, where the *identity of influenced nodes* is observed; or (3) edge semi-bandit feedback, where the *identity of influenced edges* (edges going out from influenced nodes) is observed. In this paper, we give results for the edge semi-bandit feedback model, where we observe for each influenced node, the downstream neighbors that this node influences. Such feedback is feasible to obtain in most online social networks. These networks track activities of users, for instance, when a user retweets a tweet of another user. They can thus trace the propagation (of the tweet) through the network, thereby obtaining edge semi-bandit feedback.

The IM bandits problem combines two main challenges. First, the number of actions (possible sets) $\mathcal{S}$ grows *exponentially* with the cardinality constraint on $\mathcal{S}$. Second, the agent can only observe the influenced portion of the network as feedback. Although IM bandits have been studied in the past [21, 8, 31, 5, 29] (see Section 6 for an overview and comparison), there are a number of open challenges [28]. One challenge is to identify reasonable *complexity metrics* that depend on both the topology and activation probabilities of the network and characterize the information-theoretic complexity of the IM bandits problem. Another challenge is to develop learning algorithms such that (i) their performance scales gracefully with these metrics and (ii) are computationally efficient and can be applied to large social networks with millions of users.

In this paper, we address these two challenges under the IC model with access to edge semi-bandit feedback. We refer to our model as an *independent cascade semi-bandit (ICSB)*. We make four main contributions. First, we propose `IMLinUCB`, a UCB-like algorithm for ICSBs that permits linear generalization and is suitable for large-scale problems. Second, we define a new complexity metric, referred to as *maximum observed relevance* for ICSB, which depends on the topology of the network and is a non-decreasing function of activation probabilities. The maximum observed relevance $C_*$ can also be upper bounded based on the network topology or the size of the network in the worst case. However, in real-world social networks, due to the relatively low activation probabilities [14], $C_*$ attains much smaller values as compared to the worst case upper bounds. Third, we bound the cumulative regret of `IMLinUCB`. Our regret bounds are polynomial in all quantities of interest and have near-optimal dependence on the number of interactions. They reflect the structure and activation probabilities of the network through $C_*$ and do not depend on inherently large quantities, such as the reciprocal of the minimum probability of being influenced (unlike [8]) and the cardinality of the action set. Finally, we evaluate `IMLinUCB` on several problems. Our empirical results on simple representative topologies show that the regret of `IMLinUCB` scales as suggested by our topology-dependent regret bounds. We also show that `IMLinUCB` with linear generalization can lead to low regret in real-world online influence maximization problems.

## 2 Influence Maximization under Independence Cascade Model

In this section, we define notation and give the formal problem statement for the IM problem under the IC model. Consider a directed graph $\mathcal{G} = (\mathcal{V}, \mathcal{E})$ with a set $\mathcal{V} = \{1, 2, \ldots, L\}$ of $L = |\mathcal{V}|$ nodes, a set $\mathcal{E} = \{1, 2, \ldots, |\mathcal{E}|\}$ of directed edges, and an arbitrary *binary* weight function $\mathbf{w} : \mathcal{E} \to \{0, 1\}$.

We say that a node $v_2 \in \mathcal{V}$ is *reachable* from a node $v_1 \in \mathcal{V}$ under $\mathbf{w}$ if there is a directed path[1] $p = (e_1, e_2, \ldots, e_l)$ from $v_1$ to $v_2$ in $\mathcal{G}$ satisfying $\mathbf{w}(e_i) = 1$ for all $i = 1, 2, \ldots, l$, where $e_i$ is the $i$-th edge in $p$. For a given source node set $\mathcal{S} \subseteq \mathcal{V}$ and $\mathbf{w}$, we say that node $v \in \mathcal{V}$ is *influenced* if $v$ is reachable from at least one source node in $\mathcal{S}$ under $\mathbf{w}$; and denote the number of influenced nodes in $\mathcal{G}$ by $f(\mathcal{S}, \mathbf{w})$. By definition, the nodes in $\mathcal{S}$ are always influenced.

The influence maximization (IM) problem is characterized by a triple $(\mathcal{G}, K, \overline{w})$, where $\mathcal{G}$ is a given directed graph, $K \leq L$ is the cardinality of source nodes, and $\overline{w} : \mathcal{E} \to [0, 1]$ is a probability weight function mapping each edge $e \in \mathcal{E}$ to a real number $\overline{w}(e) \in [0, 1]$. The agent needs to choose a set of $K$ source nodes $\mathcal{S} \subseteq \mathcal{V}$ based on $(\mathcal{G}, K, \overline{w})$. Then a random binary weight function $\mathbf{w}$, which encodes the diffusion process under the IC model, is obtained by independently sampling a Bernoulli random variable $\mathbf{w}(e) \sim \mathrm{Bern}\left(\overline{w}(e)\right)$ for each edge $e \in \mathcal{E}$. The agent's objective is to maximize the expected number of the influenced nodes: $\max_{\mathcal{S}: |\mathcal{S}|=K} f(\mathcal{S}, \overline{w})$, where $f(\mathcal{S}, \overline{w}) \overset{\Delta}{=} \mathbb{E}_{\mathbf{w}}\left[f(\mathcal{S}, \mathbf{w})\right]$ is the expected number of influenced nodes when the source node set is $\mathcal{S}$ and $\mathbf{w}$ is sampled according to $\overline{w}$.[2]

It is well-known that the (offline) IM problem is NP-hard [16], but can be approximately solved by approximation/randomized algorithms [6] under the IC model. In this paper, we refer to such algorithms as oracles to distinguish them from the machine learning algorithms discussed in following sections. Let $\mathcal{S}^{\mathrm{opt}}$ be the optimal solution of this problem, and $\mathcal{S}^* = \mathtt{ORACLE}(\mathcal{G}, K, \overline{w})$ be the (possibly random) solution of an oracle $\mathtt{ORACLE}$. For any $\alpha, \gamma \in [0, 1]$, we say that $\mathtt{ORACLE}$ is an $(\alpha, \gamma)$-approximation oracle for a given $(\mathcal{G}, K)$ if for any $\overline{w}$, $f(\mathcal{S}^*, \overline{w}) \geq \gamma f(\mathcal{S}^{\mathrm{opt}}, \overline{w})$ with probability at least $\alpha$. Notice that this further implies that $\mathbb{E}\left[f(\mathcal{S}^*, \overline{w})\right] \geq \alpha\gamma f(\mathcal{S}^{\mathrm{opt}}, \overline{w})$. We say an oracle is exact if $\alpha = \gamma = 1$.

# 3 Influence Maximization Semi-Bandit

In this section, we first describe the IM semi-bandit problem. Next, we state the linear generalization assumption and describe $\mathtt{IMLinUCB}$, our $\mathtt{UCB}$-based semi-bandit algorithm.

## 3.1 Protocol

The *independent cascade semi-bandit (ICSB)* problem is also characterized by a triple $(\mathcal{G}, K, \overline{w})$, but $\overline{w}$ is *unknown* to the agent. The agent interacts with the independent cascade semi-bandit for $n$ rounds. At each round $t = 1, 2, \ldots, n$, the agent first chooses a source node set $\mathcal{S}_t \subseteq \mathcal{V}$ with cardinality $K$ based on its prior information and past observations. Influence then diffuses from the nodes in $\mathcal{S}_t$ according to the IC model. Similarly to the previous section, this can be interpreted as the environment generating a binary weight function $\mathbf{w}_t$ by independently sampling $\mathbf{w}_t(e) \sim \mathrm{Bern}\left(\overline{w}(e)\right)$ for each $e \in \mathcal{E}$. At round $t$, the agent receives the reward $f(\mathcal{S}_t, \mathbf{w}_t)$, that is equal to the number of nodes influenced at that round. The agent also receives edge semi-bandit feedback from the diffusion process. Specifically, for any edge $e = (u_1, u_2) \in \mathcal{E}$, the agent observes the realization of $\mathbf{w}_t(e)$ if and only if the start node $u_1$ of the directed edge $e$ is influenced in the realization $\mathbf{w}_t$. The agent's objective is to maximize the expected cumulative reward over the $n$ steps.

## 3.2 Linear generalization

Since the number of edges in real-world social networks tends to be in millions or even billions, we need to exploit some generalization model across activation probabilities to develop efficient and deployable learning algorithms. In particular, we assume that there exists a linear-generalization model for the probability weight function $\overline{w}$. That is, each edge $e \in \mathcal{E}$ is associated with a *known* feature vector $x_e \in \Re^d$ (here $d$ is the dimension of the feature vector) and that there is an *unknown* coefficient vector $\theta^* \in \Re^d$ such that for all $e \in \mathcal{E}$, $\overline{w}(e)$ is "well approximated" by $x_e^\mathsf{T} \theta^*$. Formally, we assume that $\rho \overset{\Delta}{=} \max_{e \in \mathcal{E}} |\overline{w}(e) - x_e^\mathsf{T} \theta^*|$ is small. In Section 5.2, we see that such a linear generalization leads to efficient learning in real-world networks. Note that all vectors in this paper are column vectors.

**Algorithm 1** `IMLinUCB`: Influence Maximization Linear UCB

---

**Input:** graph $\mathcal{G}$, source node set cardinality $K$, oracle `ORACLE`, feature vector $x_e$'s, and algorithm parameters $\sigma, c > 0$,

**Initialization:** $B_0 \leftarrow 0 \in \Re^d$, $\mathbf{M}_0 \leftarrow \mathbf{I} \in \Re^{d \times d}$

**for** $t = 1, 2, \ldots, n$ **do**

    1. set $\overline{\theta}_{t-1} \leftarrow \sigma^{-2} \mathbf{M}_{t-1}^{-1} B_{t-1}$ and the UCBs as $U_t(e) \leftarrow \mathrm{Proj}_{[0,1]} \left( x_e^{\intercal} \overline{\theta}_{t-1} + c \sqrt{x_e^{\intercal} \mathbf{M}_{t-1}^{-1} x_e} \right)$

    for all $e \in \mathcal{E}$

    2. choose $\mathcal{S}_t \in$ `ORACLE`$(\mathcal{G}, K, U_t)$, and observe the edge-level semi-bandit feedback

    3. update statistics:

        (a) initialize $\mathbf{M}_t \leftarrow \mathbf{M}_{t-1}$ and $B_t \leftarrow B_{t-1}$

        (b) for all observed edges $e \in \mathcal{E}$, update $\mathbf{M}_t \leftarrow \mathbf{M}_t + \sigma^{-2} x_e x_e^{\intercal}$ and $B_t \leftarrow B_t + x_e \mathbf{w}_t(e)$

---

Similar to the existing approaches for linear bandits [1, 9], we exploit the linear generalization to develop a learning algorithm for ICSB. Without loss of generality, we assume that $\|x_e\|_2 \leq 1$ for all $e \in \mathcal{E}$. Moreover, we use $\mathbf{X} \in \Re^{|\mathcal{E}| \times d}$ to denote the feature matrix, i.e., the row of $\mathbf{X}$ associated with edge $e$ is $x_e^{\intercal}$. Note that if a learning agent does not know how to construct good features, it can always choose the naïve feature matrix $\mathbf{X} = \mathbf{I} \in \Re^{|\mathcal{E}| \times |\mathcal{E}|}$ and have no generalization model across edges. We refer to the special case $\mathbf{X} = \mathbf{I} \in \Re^{|\mathcal{E}| \times |\mathcal{E}|}$ as the *tabular* case.

### 3.3 `IMLinUCB` algorithm

In this section, we propose Influence Maximization Linear UCB (`IMLinUCB`), detailed in Algorithm 1. Notice that `IMLinUCB` represents its past observations as a positive-definite matrix (*Gram matrix*) $\mathbf{M}_t \in \Re^{d \times d}$ and a vector $B_t \in \Re^d$. Specifically, let $\mathbf{X}_t$ be a matrix whose rows are the feature vectors of all observed edges in $t$ steps and $Y_t$ be a binary column vector encoding the realizations of all observed edges in $t$ steps. Then $\mathbf{M}_t = \mathbf{I} + \sigma^{-2} \mathbf{X}_t^{\intercal} \mathbf{X}_t$ and $B_t = \mathbf{X}_t^{\intercal} Y_t$.

At each round $t$, `IMLinUCB` operates in three steps: First, it computes an upper confidence bound $U_t(e)$ for each edge $e \in \mathcal{E}$. Note that $\mathrm{Proj}_{[0,1]}(\cdot)$ projects a real number into interval $[0, 1]$ to ensure that $U_t \in [0, 1]^{|\mathcal{E}|}$. Second, it chooses a set of source nodes based on the given `ORACLE` and $U_t$, which is also a probability-weight function. Finally, it receives the edge semi-bandit feedback and uses it to update $\mathbf{M}_t$ and $B_t$. It is worth emphasizing that `IMLinUCB` is computationally efficient as long as `ORACLE` is computationally efficient. Specifically, at each round $t$, the computational complexities of both Step 1 and 3 of `IMLinUCB` are $\mathcal{O}\left(|\mathcal{E}| d^2\right)$.[3]

It is worth pointing out that in the tabular case, `IMLinUCB` reduces to `CUCB` [7], in the sense that the confidence radii in `IMLinUCB` are the same as those in `CUCB`, up to logarithmic factors. That is, `CUCB` can be viewed as a special case of `IMLinUCB` with $\mathbf{X} = \mathbf{I}$.

### 3.4 Performance metrics

Recall that the agent's objective is to maximize the expected cumulative reward, which is equivalent to minimizing the expected cumulative regret. The cumulative regret is the loss in reward (accumulated over rounds) because of the lack of knowledge of the activation probabilities. Observe that in each round $t$, `IMLinUCB` needs to use an approximation/randomized algorithm `ORACLE` for solving the offline IM problem. Naturally, this can lead to $\mathcal{O}(n)$ cumulative regret, since at each round there is a non-diminishing regret due to the approximation/randomized nature of `ORACLE`. To analyze the performance of `IMLinUCB` in such cases, we define a more appropriate performance metric, the scaled cumulative regret, as $R^{\eta}(n) = \sum_{t=1}^{n} \mathbb{E}\left[R_t^{\eta}\right]$, where $n$ is the number of steps, $\eta > 0$ is the scale, and $R_t^{\eta} = f(\mathcal{S}^{\mathrm{opt}}, \mathbf{w}_t) - \frac{1}{\eta} f(\mathcal{S}_t, \mathbf{w}_t)$ is the $\eta$-scaled realized regret $R_t^{\eta}$ at round $t$. When $\eta = 1$, $R^{\eta}(n)$ reduces to the standard expected cumulative regret $R(n)$.

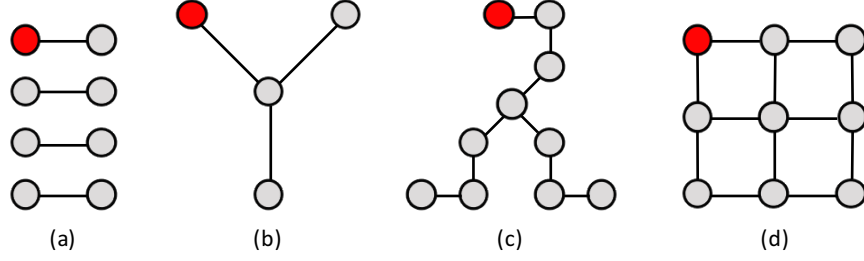

Figure 1: **a**. Bar graph on $8$ nodes. **b**. Star graph on $4$ nodes. **c**. Ray graph on $10$ nodes. **d**. Grid graph on $9$ nodes. Each undirected edge denotes two directed edges in opposite directions.

## 4   Analysis

In this section, we give a regret bound for IMLinUCB for the case when $\overline{w}(e) = x_e^\top \theta^*$ for all $e \in \mathcal{E}$, i.e., the linear generalization is perfect. Our main contribution is a regret bound that scales with a new complexity metric, *maximum observed relevance*, which depends on *both* the topology of $\mathcal{G}$ and the probability weight function $\overline{w}$, and is defined in Section 4.1. We highlight this as most known results for this problem are worst case, and some of them do not depend on probability weight function at all.

### 4.1   Maximum observed relevance

We start by defining some terminology. For given directed graph $\mathcal{G} = (\mathcal{V}, \mathcal{E})$ and source node set $\mathcal{S} \subseteq \mathcal{V}$, we say an edge $e \in \mathcal{E}$ is *relevant* to a node $v \in \mathcal{V} \setminus \mathcal{S}$ under $\mathcal{S}$ if there exists a path $p$ from a source node $s \in \mathcal{S}$ to $v$ such that (1) $e \in p$ and (2) $p$ does not contain another source node other than $s$. Notice that with a given $\mathcal{S}$, whether or not a node $v \in \mathcal{V} \setminus \mathcal{S}$ is influenced only depends on the binary weights $\mathbf{w}$ on its relevant edges. For any edge $e \in \mathcal{E}$, we define $N_{\mathcal{S},e}$ as the number of nodes in $\mathcal{V} \setminus \mathcal{S}$ it is relevant to, and define $P_{\mathcal{S},e}$ as the conditional probability that $e$ is observed given $\mathcal{S}$,

$$N_{\mathcal{S},e} \triangleq \sum_{v \in \mathcal{V} \setminus \mathcal{S}} \mathbf{1}\{e \text{ is relevant to } v \text{ under } \mathcal{S}\} \quad \text{and} \quad P_{\mathcal{S},e} \triangleq \mathbb{P}\left(e \text{ is observed} \mid \mathcal{S}\right). \quad (1)$$

Notice that $N_{\mathcal{S},e}$ only depends on the topology of $\mathcal{G}$, while $P_{\mathcal{S},e}$ depends on *both* the topology of $\mathcal{G}$ and the probability weight $\overline{w}$. The *maximum observed relevance* $C_*$ is defined as the maximum (over $\mathcal{S}$) 2-norm of $N_{\mathcal{S},e}$'s weighted by $P_{\mathcal{S},e}$'s,

$$C_* \triangleq \max_{\mathcal{S}:\, |\mathcal{S}|=K} \sqrt{\sum_{e \in \mathcal{E}} N_{\mathcal{S},e}^2 P_{\mathcal{S},e}}. \quad (2)$$

As is detailed in the proof of Lemma 1 in Appendix A, $C_*$ arises in the step where Cauchy-Schwarz inequality is applied. Note that $C_*$ also depends on both the topology of $\mathcal{G}$ and the probability weight $\overline{w}$. However, $C_*$ can be bounded from above only based on the topology of $\mathcal{G}$ or the size of the problem, i.e., $L = |\mathcal{V}|$ and $|\mathcal{E}|$. Specifically, by defining $C_{\mathcal{G}} \triangleq \max_{\mathcal{S}:\, |\mathcal{S}|=K} \sqrt{\sum_{e \in \mathcal{E}} N_{\mathcal{S},e}^2}$, we have

$$C_* \le C_{\mathcal{G}} = \max_{\mathcal{S}:\, |\mathcal{S}|=K} \sqrt{\sum_{e \in \mathcal{E}} N_{\mathcal{S},e}^2} \le (L-K)\sqrt{|\mathcal{E}|} = \mathcal{O}\left(L\sqrt{|\mathcal{E}|}\right) = \mathcal{O}\left(L^2\right), \quad (3)$$

where $C_{\mathcal{G}}$ is the maximum/worst-case (over $\overline{w}$) $C_*$ for the directed graph $\mathcal{G}$, and the maximum is obtained by setting $\overline{w}(e) = 1$ for all $e \in \mathcal{E}$. Since $C_{\mathcal{G}}$ is worst-case, it might be very far away from $C_*$ if the activation probabilities are small. Indeed, this is what we expect in typical real-world situations. Notice also that if $\max_{e \in \mathcal{E}} \overline{w}(e) \to 0$, then $P_{\mathcal{S},e} \to 0$ for all $e \notin \mathcal{E}(\mathcal{S})$ and $P_{\mathcal{S},e} = 1$ for all $e \in \mathcal{E}(\mathcal{S})$, where $\mathcal{E}(\mathcal{S})$ is the set of edges with start node in $\mathcal{S}$, hence we have $C_* \to C_{\mathcal{G}}^0 \triangleq \max_{\mathcal{S}:\, |\mathcal{S}|=K} \sqrt{\sum_{e \in \mathcal{E}(\mathcal{S})} N_{\mathcal{S},e}^2}$. In particular, if $K$ is small, $C_{\mathcal{G}}^0$ is much less than $C_{\mathcal{G}}$ in many topologies. For example, in a complete graph with $K=1$, $C_{\mathcal{G}} = \Theta(L^2)$ while $C_{\mathcal{G}}^0 = \Theta(L^{\frac{3}{2}})$. Finally, it is worth pointing out that there exist situations $(\mathcal{G}, \overline{w})$ such that $C_* = \Theta(L^2)$. One such example is when $\mathcal{G}$ is a complete graph with $L$ nodes and $\overline{w}(e) = L/(L+1)$ for all edges $e$ in this graph.

To give more intuition, in the rest of this subsection, we illustrate how $C_{\mathcal{G}}$, the *worst-case* $C_*$, varies with four graph topologies in Figure 1: bar, star, ray, and grid, as well as two other topologies:

general tree and complete graph. We fix the node set $\mathcal{V} = \{1, 2, \ldots, L\}$ for all graphs. The bar graph (Figure 1a) is a graph where nodes $i$ and $i + 1$ are connected when $i$ is odd. The star graph (Figure 1b) is a graph where node 1 is central and all remaining nodes $i \in \mathcal{V} \setminus \{1\}$ are connected to it. The distance between any two of these nodes is 2. The ray graph (Figure 1c) is a star graph with $k = \lceil \sqrt{L-1} \rceil$ arms, where node 1 is central and each arm contains either $\lceil (L-1)/k \rceil$ or $\lfloor (L-1)/k \rfloor$ nodes connected in a line. The distance between any two nodes in this graph is $\mathcal{O}(\sqrt{L})$. The grid graph (Figure 1d) is a classical non-tree graph with $\mathcal{O}(L)$ edges.

To see how $C_{\mathcal{G}}$ varies with the graph topology, we start with the simplified case when $K = |\mathcal{S}| = 1$. In the bar graph (Figure 1a), only one edge is relevant to a node $v \in \mathcal{V} \setminus \mathcal{S}$ and all the other edges are not relevant to any nodes. Therefore, $C_{\mathcal{G}} \leq 1$. In the star graph (Figure 1b), for any $s$, at most one edge is relevant to at most $L - 1$ nodes and the remaining edges are relevant to at most one node. In this case, $C_{\mathcal{G}} \leq \sqrt{L^2 + L} = \mathcal{O}(L)$. In the ray graph (Figure 1c), for any $s$, at most $\mathcal{O}(\sqrt{L})$ edges are relevant to $L - 1$ nodes and the remaining edges are relevant to at most $\mathcal{O}(\sqrt{L})$ nodes. In this case, $C_{\mathcal{G}} = \mathcal{O}(\sqrt{L^{\frac{1}{2}} L^2 + LL}) = \mathcal{O}(L^{\frac{5}{4}})$. Finally, recall that for all graphs we can bound $C_{\mathcal{G}}$ by $\mathcal{O}(L\sqrt{|\mathcal{E}|})$, regardless of $K$. Hence, for the grid graph (Figure 1d) and general tree graph, $C_{\mathcal{G}} = \mathcal{O}(L^{\frac{3}{2}})$ since $|\mathcal{E}| = \mathcal{O}(L)$; for the complete graph $C_{\mathcal{G}} = \mathcal{O}(L^2)$ since $|\mathcal{E}| = \mathcal{O}(L^2)$. Clearly, $C_{\mathcal{G}}$ varies widely with the topology of the graph. The second column of Table 1 summarizes how $C_{\mathcal{G}}$ varies with the above-mentioned graph topologies for general $K = |\mathcal{S}|$.

## 4.2 Regret guarantees

Consider $C_*$ defined in Section 4.1 and recall the worst-case upper bound $C_* \leq (L - K)\sqrt{|\mathcal{E}|}$, we have the following regret guarantees for `IMLinUCB`.

**Theorem 1** *Assume that (1) $\overline{w}(e) = x_e^\intercal \theta^*$ for all $e \in \mathcal{E}$ and (2) `ORACLE` is an $(\alpha, \gamma)$-approximation algorithm. Let $D$ be a known upper bound on $\|\theta^*\|_2$, if we apply `IMLinUCB` with $\sigma = 1$ and*

$$c = \sqrt{d \log \left(1 + \frac{n|\mathcal{E}|}{d}\right) + 2\log\left(n(L + 1 - K)\right)} + D, \tag{4}$$

*then we have*

$$R^{\alpha\gamma}(n) \leq \frac{2cC_*}{\alpha\gamma} \sqrt{dn|\mathcal{E}| \log_2\left(1 + \frac{n|\mathcal{E}|}{d}\right)} + 1 = \widetilde{\mathcal{O}}\left(dC_*\sqrt{|\mathcal{E}|n}/(\alpha\gamma)\right) \tag{5}$$

$$\leq \widetilde{\mathcal{O}}\left(d(L - K)|\mathcal{E}|\sqrt{n}/(\alpha\gamma)\right). \tag{6}$$

*Moreover, if the feature matrix $\mathbf{X} = \mathbf{I} \in \Re^{|\mathcal{E}| \times |\mathcal{E}|}$ (i.e., the tabular case), we have*

$$R^{\alpha\gamma}(n) \leq \frac{2cC_*}{\alpha\gamma} \sqrt{n|\mathcal{E}| \log_2(1 + n)} + 1 = \widetilde{\mathcal{O}}\left(|\mathcal{E}|C_*\sqrt{n}/(\alpha\gamma)\right) \tag{7}$$

$$\leq \widetilde{\mathcal{O}}\left((L - K)|\mathcal{E}|^{\frac{3}{2}}\sqrt{n}/(\alpha\gamma)\right). \tag{8}$$

Please refer to Appendix A for the proof of Theorem 1, that we outline in Section 4.3. We now briefly comment on the regret bounds in Theorem 1.

**Topology-dependent bounds:** Since $C_*$ is topology-dependent, the regret bounds in Equations 5 and 7 are also topology-dependent. Table 1 summarizes the regret bounds for each topology[4] discussed in Section 4.1. Since the regret bounds in Table 1 are the worst-case regret bounds for a given topology, more general topologies have larger regret bounds. For instance, the regret bounds for tree are larger than their counterparts for star and ray, since star and ray are special trees. The grid and tree can also be viewed as special complete graphs by setting $\overline{w}(e) = 0$ for some $e \in \mathcal{E}$, hence complete graph has larger regret bounds. Again, in practice we expect $C_*$ to be far smaller due to activation probabilities.

| topology | $C_{\mathcal{G}}$ (worst-case $C_*$) | $R^{\alpha\gamma}(n)$ for general $\mathbf{X}$ | $R^{\alpha\gamma}(n)$ for $\mathbf{X} = \mathbf{I}$ |
|---|---|---|---|
| bar graph | $\mathcal{O}(\sqrt{K})$ | $\widetilde{\mathcal{O}}\left(dK\sqrt{n}/(\alpha\gamma)\right)$ | $\widetilde{\mathcal{O}}\left(L\sqrt{Kn}/(\alpha\gamma)\right)$ |
| star graph | $\mathcal{O}(L\sqrt{K})$ | $\widetilde{\mathcal{O}}\left(dL^{\frac{3}{2}}\sqrt{Kn}/(\alpha\gamma)\right)$ | $\widetilde{\mathcal{O}}\left(L^2\sqrt{Kn}/(\alpha\gamma)\right)$ |
| ray graph | $\mathcal{O}(L^{\frac{5}{4}}\sqrt{K})$ | $\widetilde{\mathcal{O}}\left(dL^{\frac{7}{4}}\sqrt{Kn}/(\alpha\gamma)\right)$ | $\widetilde{\mathcal{O}}\left(L^{\frac{9}{4}}\sqrt{Kn}/(\alpha\gamma)\right)$ |
| tree graph | $\mathcal{O}(L^{\frac{3}{2}})$ | $\widetilde{\mathcal{O}}\left(dL^2\sqrt{n}/(\alpha\gamma)\right)$ | $\widetilde{\mathcal{O}}\left(L^{\frac{5}{2}}\sqrt{n}/(\alpha\gamma)\right)$ |
| grid graph | $\mathcal{O}(L^{\frac{3}{2}})$ | $\widetilde{\mathcal{O}}\left(dL^2\sqrt{n}/(\alpha\gamma)\right)$ | $\widetilde{\mathcal{O}}\left(L^{\frac{5}{2}}\sqrt{n}/(\alpha\gamma)\right)$ |
| complete graph | $\mathcal{O}(L^2)$ | $\widetilde{\mathcal{O}}\left(dL^3\sqrt{n}/(\alpha\gamma)\right)$ | $\widetilde{\mathcal{O}}\left(L^4\sqrt{n}/(\alpha\gamma)\right)$ |

Table 1: $C_{\mathcal{G}}$ and *worst-case* regret bounds for different graph topologies.

**Tighter bounds in tabular case and under exact oracle:** Notice that for the tabular case with feature matrix $\mathbf{X} = \mathbf{I}$ and $d = |\mathcal{E}|$, $\widetilde{\mathcal{O}}(\sqrt{|\mathcal{E}|})$ tighter regret bounds are obtained in Equations 7 and 8. Also notice that the $\widetilde{\mathcal{O}}(1/(\alpha\gamma))$ factor is due to the fact that ORACLE is an $(\alpha, \gamma)$-approximation oracle. If ORACLE solves the IM problem exactly (i.e., $\alpha = \gamma = 1$), then $R^{\alpha\gamma}(n) = R(n)$.

**Tightness of our regret bounds:** First, note that our regret bound in the bar case with $K = 1$ matches the regret bound of the classic LinUCB algorithm. Specifically, with perfect linear generalization, this case is equivalent to a linear bandit problem with $L$ arms and feature dimension $d$. From Table 1, our regret bound in this case is $\widetilde{\mathcal{O}}(d\sqrt{n})$, which matches the known regret bound of LinUCB that can be obtained by the technique of [1]. Second, we briefly discuss the tightness of the regret bound in Equation 6 for a general graph with $L$ nodes and $|\mathcal{E}|$ edges. Note that the $\widetilde{\mathcal{O}}(\sqrt{n})$-dependence on time is near-optimal, and the $\widetilde{\mathcal{O}}(d)$-dependence on feature dimension is standard in linear bandits [1, 33], since $\widetilde{\mathcal{O}}(\sqrt{d})$ results are only known for impractical algorithms. The $\widetilde{\mathcal{O}}(L - K)$ factor is due to the fact that the reward in this problem is from $K$ to $L$, rather than from 0 to 1. To explain the $\widetilde{\mathcal{O}}(|\mathcal{E}|)$ factor in this bound, notice that one $\widetilde{\mathcal{O}}(\sqrt{|\mathcal{E}|})$ factor is due to the fact that at most $\widetilde{\mathcal{O}}(|\mathcal{E}|)$ edges might be observed at each round (see Theorem 3), and is intrinsic to the problem similarly to combinatorial semi-bandits [19]; another $\widetilde{\mathcal{O}}(\sqrt{|\mathcal{E}|})$ factor is due to linear generalization (see Lemma 1) and might be removed by better analysis. We conjecture that our $\widetilde{\mathcal{O}}\left(d(L - K)|\mathcal{E}|\sqrt{n}/(\alpha\gamma)\right)$ regret bound in this case is at most $\widetilde{\mathcal{O}}(\sqrt{|\mathcal{E}|d})$ away from being tight.

### 4.3 Proof sketch

We now outline the proof of Theorem 1. For each round $t \leq n$, we define the favorable event $\xi_{t-1} = \{|x_e^\top(\overline{\theta}_{\tau-1} - \theta^*)| \leq c\sqrt{x_e^\top \mathbf{M}_{\tau-1}^{-1} x_e}, \forall e \in \mathcal{E}, \forall \tau \leq t\}$, and the unfavorable event $\overline{\xi}_{t-1}$ as the complement of $\xi_{t-1}$. If we decompose $\mathbb{E}[R_t^{\alpha\gamma}]$, the $(\alpha\gamma)$-scaled expected regret at round $t$, over events $\xi_{t-1}$ and $\overline{\xi}_{t-1}$, and bound $R_t^{\alpha\gamma}$ on event $\overline{\xi}_{t-1}$ using the naïve bound $R_t^{\alpha\gamma} \leq L - K$, then,

$$\mathbb{E}[R_t^{\alpha\gamma}] \leq \mathbb{P}(\xi_{t-1})\,\mathbb{E}\left[R_t^{\alpha\gamma}|\xi_{t-1}\right] + \mathbb{P}\left(\overline{\xi}_{t-1}\right)[L - K].$$

By choosing $c$ as specified by Equation 4, we have $\mathbb{P}\left(\overline{\xi}_{t-1}\right)[L - K] < 1/n$ (see Lemma 2 in the appendix). On the other hand, notice that by definition of $\xi_{t-1}$, $\overline{w}(e) \leq U_t(e), \forall e \in \mathcal{E}$ under event $\xi_{t-1}$. Using the monotonicity of $f$ in the probability weight, and the fact that ORACLE is an $(\alpha, \gamma)$-approximation algorithm, we have

$$\mathbb{E}\left[R_t^{\alpha\gamma}|\xi_{t-1}\right] \leq \mathbb{E}\left[f(\mathcal{S}_t, U_t) - f(\mathcal{S}_t, \overline{w})|\xi_{t-1}\right]/(\alpha\gamma).$$

The next observation is that, from the linearity of expectation, the gap $f(\mathcal{S}_t, U_t) - f(\mathcal{S}_t, \overline{w})$ decomposes over nodes $v \in \mathcal{V} \setminus \mathcal{S}_t$. Specifically, for any source node set $\mathcal{S} \subseteq \mathcal{V}$, any probability weight function $w : \mathcal{E} \to [0, 1]$, and any node $v \in \mathcal{V}$, we define $f(\mathcal{S}, w, v)$ as the probability that node $v$ is influenced if the source node set is $\mathcal{S}$ and the probability weight is $w$. Hence, we have

$$f(\mathcal{S}_t, U_t) - f(\mathcal{S}_t, \overline{w}) = \sum_{v \in \mathcal{V} \setminus \mathcal{S}_t}\left[f(\mathcal{S}_t, U_t, v) - f(\mathcal{S}_t, \overline{w}, v)\right].$$

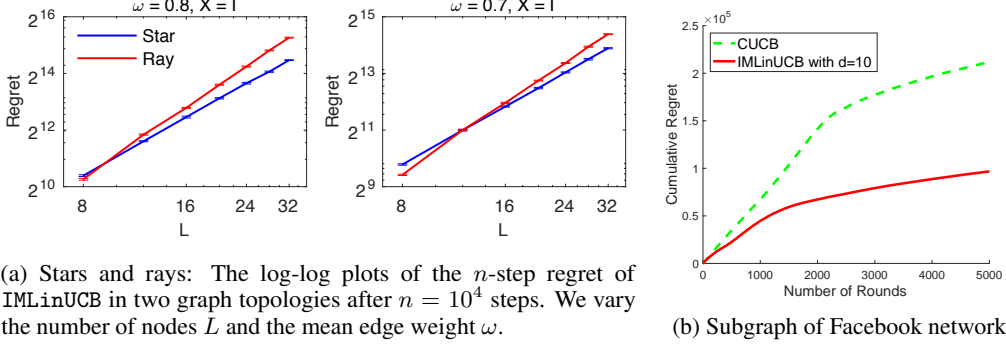

(a) Stars and rays: The log-log plots of the $n$-step regret of `IMLinUCB` in two graph topologies after $n = 10^4$ steps. We vary the number of nodes $L$ and the mean edge weight $\omega$.

(b) Subgraph of Facebook network

Figure 2: Experimental results

In the appendix, we show that under any weight function, the diffusion process from the source node set $S_t$ to the target node $v$ can be modeled as a Markov chain. Hence, weight function $U_t$ and $\overline{w}$ give us two Markov chains with the same state space but different transition probabilities. $f(S_t, U_t, v) - f(S_t, \overline{w}, v)$ can be recursively bounded based on the state diagram of the Markov chain under weight function $\overline{w}$. With some algebra, Theorem 3 in Appendix A bounds $f(S_t, U_t, v) - f(S_t, \overline{w}, v)$ by the edge-level gap $U_t(e) - \overline{w}(e)$ on the observed relevant edges for node $v$,

$$f(\mathcal{S}_t, U_t, v) - f(\mathcal{S}_t, \overline{w}, v) \leq \sum_{e \in \mathcal{E}_{\mathcal{S}_t, v}} \mathbb{E}\left[\mathbf{1}\left\{O_t(e)\right\}[U_t(e) - \overline{w}(e)]|\mathcal{H}_{t-1}, \mathcal{S}_t\right], \quad (9)$$

for any $t$, any "history" (past observations) $\mathcal{H}_{t-1}$ and $\mathcal{S}_t$ such that $\xi_{t-1}$ holds, and any $v \in \mathcal{V} \setminus \mathcal{S}_t$, where $\mathcal{E}_{\mathcal{S}_t, v}$ is the set of edges relevant to $v$ and $O_t(e)$ is the event that edge $e$ is observed at round $t$. Based on Equation 9, we can prove Theorem 1 using the standard linear-bandit techniques (see Appendix A).

## 5 Experiments

In this section, we present a synthetic experiment in order to empirically validate our upper bounds on the regret. Next, we evaluate our algorithm on a real-world Facebook subgraph.

### 5.1 Stars and rays

In the first experiment, we evaluate `IMLinUCB` on undirected stars and rays (Figure 1) and validate that the regret grows with the number of nodes $L$ and the maximum observed relevance $C_*$ as shown in Table 1. We focus on the tabular case ($\mathbf{X} = \mathbf{I}$) with $K = |\mathcal{S}| = 1$, where the IM problem can be solved exactly. We vary the number of nodes $L$; and edge weight $\overline{w}(e) = \omega$, which is the same for all edges $e$. We run `IMLinUCB` for $n = 10^4$ steps and verify that it converges to the optimal solution in each experiment. We report the $n$-step regret of `IMLinUCB` for $8 \leq L \leq 32$ in Figure 2a. Recall that from Table 1, $R(n) = \widetilde{\mathcal{O}}(L^2)$ for star and $R(n) = \widetilde{\mathcal{O}}(L^{\frac{9}{4}})$ for ray.

We numerically estimate the growth of regret in $L$, the exponent of $L$, in the log-log space of $L$ and regret. In particular, since $\log(f(L)) = p \log(L) + \log(c)$ for any $f(L) = cL^p$ and $c > 0$, both $p$ and $\log(c)$ can be estimated by linear regression in the new space. For star graphs with $\omega = 0.8$ and $\omega = 0.7$, our estimated growth are respectively $\mathcal{O}(L^{2.040})$ and $\mathcal{O}(L^{2.056})$, which are close to the expected $\widetilde{\mathcal{O}}(L^2)$. For ray graphs with $\omega = 0.8$ and $\omega = 0.7$, our estimated growth are respectively $\mathcal{O}(L^{2.488})$ and $\mathcal{O}(L^{2.467})$, which are again close to the expected $\widetilde{\mathcal{O}}(L^{\frac{9}{4}})$. This shows that maximum observed relevance $C_*$ proposed in Section 4.1 is a reasonable complexity metric for these two topologies.

### 5.2 Subgraph of Facebook network

In the second experiment, we demonstrate the potential performance gain of `IMLinUCB` in real-world influence maximization semi-bandit problems by exploiting linear generalization across edges. Specifically, we compare `IMLinUCB` with `CUCB` in a subgraph of Facebook network from [22]. The subgraph has $L = |\mathcal{V}| = 327$ nodes and $|\mathcal{E}| = 5038$ directed edges. Since the true probability weight

function $\overline{w}$ is not available, we independently sample $\overline{w}(e)$'s from the uniform distribution $U(0, 0.1)$ and treat them as ground-truth. Note that this range of probabilities is guided by empirical evidence in [14, 3]. We set $n = 5000$ and $K = 10$ in this experiment. For `IMLinUCB`, we choose $d = 10$ and generate edge feature $x_e$'s as follows: we first use `node2vec` algorithm [15] to generate a node feature in $\Re^d$ for each node $v \in \mathcal{V}$; then for each edge $e$, we generate $x_e$ as the element-wise product of node features of the two nodes connected to $e$. Note that the linear generalization in this experiment is imperfect in the sense that $\min_{\theta \in \Re^d} \max_{e \in \mathcal{E}} |\overline{w}(e) - x_e^T \theta| > 0$. For both `CUCB` and `IMLinUCB`, we choose `ORACLE` as the state-of-the-art offline IM algorithm proposed in [27]. To compute the cumulative regret, we compare against a fixed seed set $\mathcal{S}^*$ obtained by using the true $\overline{w}$ as input to the oracle proposed in [27]. We average the empirical cumulative regret over 10 independent runs, and plot the results in Figure 2b. The experimental results show that compared with `CUCB`, `IMLinUCB` can significantly reduce the cumulative regret by exploiting linear generalization across $\overline{w}(e)$'s.

## 6 Related Work

There exist prior results on IM semi-bandits [21, 8, 31]. First, Lei *et al.* [21] gave algorithms for the same feedback model as ours. The algorithms are not analyzed and cannot solve large-scale problems because they estimate each edge weight independently. Second, our setting is a special case of stochastic combinatorial semi-bandit with a submodular reward function and stochastically observed edges [8]. Their work is the closest related work. Their gap-dependent and gap-free bounds are both problematic because they depend on the reciprocal of the minimum observation probability $p^*$ of an edge: Consider a line graph with $|\mathcal{E}|$ edges where all edge weights are $0.5$. Then $1/p^*$ is $2^{|\mathcal{E}|-1}$. On the other hand, our derived regret bounds in Theorem 1 are polynomial in all quantities of interest. A very recent result of Wang and Chen [32] removes the $1/p^*$ factor in [8] for the tabular case and presents a worst-case bound of $\widetilde{\mathcal{O}}(L|\mathcal{E}|\sqrt{n})$, which in the tabular complete graph case improves over our result by $\widetilde{\mathcal{O}}(L)$. On the other hand, their analysis does not give structural guarantees that we provide with maximum observed relevance $C_*$ obtaining potentially much better results for the case in hand and giving insights for the complexity of IM bandits. Moreover, both Chen *et al.* [8] and Wang and Chen [32] do not consider generalization models across edges or nodes, and therefore their proposed algorithms are unlikely to be practical for real-world social networks. In contrast, our proposed algorithm scales to large problems by exploiting linear generalization across edges.

**IM bandits for different influence models and settings:** There exist a number of extensions and related results for IM bandits. We only mention the most related ones (see [28] for a recent survey). Vaswani *et al.* [31] proposed a learning algorithm for a different and more challenging feedback model, where the learning agent observes influenced *nodes but not the edges*, but they do not give any guarantees. Carpentier and Valko [5] give a minimax optimal algorithm for IM bandits but only consider a *local model* of influence with a *single* source and a cascade of influences never happens. In related networked bandits [11], the learner chooses a node and its reward is the *sum* of the rewards of the chosen node and its neighborhood. The problem gets more challenging when we allow the influence probabilities to change [2], when we allow the seed set to be chosen adaptively [30], or when we consider a continuous model [12]. Furthermore, Sigla *et al.* [26] treats the IM setting with an additional observability constraints, where we face a restriction on which nodes we can choose at each round. This setting is also related to the *volatile multi-armed bandits* where the set of possible arms changes [4]. Vaswani *et al.* [29] proposed a diffusion-independent algorithm for IM semi-bandits with a wide range of diffusion models, based on the maximum-reachability approximation. Despite its wide applicability, the maximum reachability approximation introduces an additional approximation factor to the scaled regret bounds. As they have discussed, this approximation factor can be large in some cases. Lagrée *et al.* [20] treat a *persistent* extension of IM bandits when some nodes become persistent over the rounds and no longer yield rewards. This work is also a generalization and extension of recent work on cascading bandits [17, 18, 34], since cascading bandits can be viewed as variants of online influence maximization problems with special topologies (chains).

**Acknowledgements** The research presented was supported by French Ministry of Higher Education and Research, Nord-Pas-de-Calais Regional Council and French National Research Agency projects ExTra-Learn (n.ANR-14-CE24-0010-01) and BoB (n.ANR-16-CE23-0003). We would also like to thank Dr. Wei Chen and Mr. Qinshi Wang for pointing out a mistake in an earlier version of this paper.

## Footnotes

[1]As is standard in graph theory, a directed path is a sequence of directed edges connecting a sequence of distinct nodes, under the restriction that all edges are directed in the same direction.

[2]Notice that the definitions of $f(\mathcal{S}, \overline{w})$ and $f(\mathcal{S}, \mathbf{w})$ are consistent in the sense that if $\overline{w} \in \{0, 1\}^{|\mathcal{E}|}$, then $f(\mathcal{S}, \overline{w}) = f(\mathcal{S}, \mathbf{w})$ with probability 1.

[3]Notice that in a practical implementation, we store $\mathbf{M}_t^{-1}$ instead of $\mathbf{M}_t$. Moreover, $\mathbf{M}_t \leftarrow \mathbf{M}_t + \sigma^{-2} x_e x_e^{\intercal}$ is equivalent to $\mathbf{M}_t^{-1} \leftarrow \mathbf{M}_t^{-1} - \frac{\mathbf{M}_t^{-1} x_e x_e^{\intercal} \mathbf{M}_t^{-1}}{x_e^{\intercal} \mathbf{M}_t^{-1} x_e + \sigma^2}$.

[4]The regret bound for bar graph is based on Theorem 2 in the appendix, which is a stronger version of Theorem 1 for disconnected graph.

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
