[Supplementary Material · OIM_Appendix.pdf]

# Appendix

## A    Proof of Theorem 1

In the appendix, we prove a slightly stronger version of Theorem 1, which also uses another complexity metric $E_*$ defined as follows: Assume that the graph $\mathcal{G} = (\mathcal{V}, \mathcal{E})$ includes $m$ disconnected subgraphs $\mathcal{G}_1 = (\mathcal{V}_1, \mathcal{E}_1), \mathcal{G}_2 = (\mathcal{V}_2, \mathcal{E}_2), \ldots, \mathcal{G}_m = (\mathcal{V}_m, \mathcal{E}_m)$, which are in the descending order based on the number of nodes $|\mathcal{E}_i|$'s. We define $E_*$ as the number of edges in the first $\min\{m, K\}$ subgraphs:

$$E_* = \sum_{i=1}^{\min\{m,K\}} |\mathcal{E}_i|. \tag{10}$$

Note that by definition, $E_* \le |\mathcal{E}|$. Based on $E_*$, we have the following slightly stronger version of Theorem 1.

**Theorem 2** *Assume that (1) $\overline{w}(e) = x_e^\mathsf{T} \theta^*$ for all $e \in \mathcal{E}$ and (2) $\mathtt{ORACLE}$ is an $(\alpha, \gamma)$-approximation algorithm. Let $D$ be a known upper bound on $\|\theta^*\|_2$. If we apply $\mathtt{IMLinUCB}$ with $\sigma = 1$ and*

$$c \ge \sqrt{d \log\left(1 + \frac{nE_*}{d}\right) + 2\log\left(n(L + 1 - K)\right)} + D, \tag{11}$$

*then we have*

$$R^{\alpha\gamma}(n) \le \frac{2cC_*}{\alpha\gamma} \sqrt{dnE_* \log_2\left(1 + \frac{nE_*}{d}\right)} + 1 = \widetilde{\mathcal{O}}\left(dC_* \sqrt{E_* n}/(\alpha\gamma)\right). \tag{12}$$

*Moreover, if the feature matrix is of the form $X = I \in \Re^{|\mathcal{E}| \times |\mathcal{E}|}$ (i.e., the tabular case), we have*

$$R^{\alpha\gamma}(n) \le \frac{2cC_*}{\alpha\gamma} \sqrt{n|\mathcal{E}| \log_2(1 + n)} + 1 = \widetilde{\mathcal{O}}\left(|\mathcal{E}|C_* \sqrt{n}/(\alpha\gamma)\right). \tag{13}$$

Since $E_* \le |\mathcal{E}|$, Theorem 2 implies Theorem 1. We prove Theorem 2 in the remainder of this section.

We now define some notation to simplify the exposition throughout this section.

**Definition 1** *For any source node set $\mathcal{S} \subseteq \mathcal{V}$, any probability weight function $w : \mathcal{E} \to [0, 1]$, and any node $v \in \mathcal{V}$, we define $f(\mathcal{S}, w, v)$ as the probability that node $v$ is influenced if the source node set is $\mathcal{S}$ and the probability weight function is $w$.*

Notice that by definition, $f(\mathcal{S}, w) = \sum_{v \in \mathcal{V}} f(\mathcal{S}, w, v)$ always holds. Moreover, if $v \in \mathcal{S}$, then $f(\mathcal{S}, w, v) = 1$ for any $w$ by the definition of the influence model.

**Definition 2** *For any round $t$ and any directed edge $e \in \mathcal{E}$, we define event*

$$O_t(e) = \{\text{edge } e \text{ is observed at round } t\}.$$

Note that by definition, an directed edge $e$ is observed if and only if its start node is influenced and observed does not necessarily mean that the edge is *active*.

### A.1    Proof of Theorem 2

**Proof:**    Let $\mathcal{H}_t$ be the history ($\sigma$-algebra) of past observations and actions by the end of round $t$. By the definition of $R_t^{\alpha\gamma}$, we have

$$\mathbb{E}\left[R_t^{\alpha\gamma} | \mathcal{H}_{t-1}\right] = f(\mathcal{S}^{\mathrm{opt}}, \overline{w}) - \frac{1}{\alpha\gamma} \mathbb{E}\left[f(\mathcal{S}_t, \overline{w}) | \mathcal{H}_{t-1}\right], \tag{14}$$

where the expectation is over the possible randomness of $\mathcal{S}_t$, since $\mathtt{ORACLE}$ might be a randomized algorithm. Notice that the randomness coming from the edge activation is already taken care of in the

definition of $f$. For any $t \leq n$, we define event $\xi_{t-1}$ as

$$\xi_{t-1} = \left\{ |x_e^\intercal(\overline{\theta}_{\tau-1} - \theta^*)| \leq c\sqrt{x_e^\intercal \mathbf{M}_{\tau-1}^{-1} x_e}, \forall e \in \mathcal{E}, \forall \tau \leq t \right\}, \tag{15}$$

and $\overline{\xi}_{t-1}$ as the complement of $\xi_{t-1}$. Notice that $\xi_{t-1}$ is $\mathcal{H}_{t-1}$-measurable. Hence we have

$$\mathbb{E}[R_t^{\alpha\gamma}] \leq \mathbb{P}(\xi_{t-1}) \mathbb{E}\left[f(\mathcal{S}^{\mathrm{opt}}, \overline{w}) - f(\mathcal{S}_t, \overline{w})/(\alpha\gamma)|\xi_{t-1}\right] + \mathbb{P}(\overline{\xi}_{t-1}) [L - K].$$

Notice that under event $\xi_{t-1}$, $\overline{w}(e) \leq U_t(e)$, $\forall e \in \mathcal{E}$, for all $t \leq n$, thus we have

$$f(\mathcal{S}^{\mathrm{opt}}, \overline{w}) \leq f(\mathcal{S}^{\mathrm{opt}}, U_t) \leq \max_{\mathcal{S}: |\mathcal{S}|=K} f(\mathcal{S}, U_t) \leq \frac{1}{\alpha\gamma} \mathbb{E}\left[f(\mathcal{S}_t, U_t)| \mathcal{H}_{t-1}\right],$$

where the first inequality follows from the monotonicity of $f$ in the probability weight, and the last inequality follows from the fact that ORACLE is an $(\alpha, \gamma)$-approximation algorithm. Thus, we have

$$\mathbb{E}[R_t^{\alpha\gamma}] \leq \frac{\mathbb{P}(\xi_{t-1})}{\alpha\gamma} \mathbb{E}\left[f(\mathcal{S}_t, U_t) - f(\mathcal{S}_t, \overline{w})|\xi_{t-1}\right] + \mathbb{P}(\overline{\xi}_{t-1}) [L - K]. \tag{16}$$

Notice that based on Definition 1, we have

$$f(\mathcal{S}_t, U_t) - f(\mathcal{S}_t, \overline{w}) = \sum_{v \in \mathcal{V} \setminus \mathcal{S}_t} \left[f(\mathcal{S}_t, U_t, v) - f(\mathcal{S}_t, \overline{w}, v)\right].$$

Recall that for a given graph $\mathcal{G} = (\mathcal{V}, \mathcal{E})$ and a given source node set $\mathcal{S} \subseteq \mathcal{V}$, we say an edge $e \in \mathcal{E}$ and a node $v \in \mathcal{V} \setminus \mathcal{S}$ are *relevant* if there exists a path $p$ from a source node $s \in \mathcal{S}$ to $v$ such that (1) $e \in p$ and (2) $p$ does not contain another source node other than $s$. We use $\mathcal{E}_{\mathcal{S},v} \subseteq \mathcal{E}$ to denote the set of edges relevant to node $v$ under the source node set $\mathcal{S}$, and use $\mathcal{V}_{\mathcal{S},v} \subseteq \mathcal{V}$ to denote the set of nodes connected to at least one edge in $\mathcal{E}_{\mathcal{S},v}$. Notice that $\mathcal{G}_{\mathcal{S},v} \triangleq (\mathcal{V}_{\mathcal{S},v}, \mathcal{E}_{\mathcal{S},v})$ is a subgraph of $\mathcal{G}$, and we refer to it as the **relevant subgraph** of node $v$ under the source node set $\mathcal{S}$.

Based on the notion of relevant subgraph, we have the following theorem, which bounds $f(\mathcal{S}_t, U_t, v) - f(\mathcal{S}_t, \overline{w}, v)$ by edge-level gaps $U_t(e) - \overline{w}(e)$ on the observed edges in the relevant subgraph $\mathcal{G}_{\mathcal{S}_t,v}$ for node $v$;

**Theorem 3** *For any $t$, any history $\mathcal{H}_{t-1}$ and $\mathcal{S}_t$ such that $\xi_{t-1}$ holds, and any $v \in \mathcal{V} \setminus \mathcal{S}_t$, we have*

$$f(\mathcal{S}_t, U_t, v) - f(\mathcal{S}_t, \overline{w}, v) \leq \sum_{e \in \mathcal{E}_{\mathcal{S}_t,v}} \mathbb{E}\left[\mathbf{1}\left\{O_t(e)\right\} [U_t(e) - \overline{w}(e)]|\mathcal{H}_{t-1}, \mathcal{S}_t\right],$$

*where $\mathcal{E}_{\mathcal{S}_t,v}$ is the edge set of the relevant subgraph $\mathcal{G}_{\mathcal{S}_t,v}$.*

Please refer to Section A.2 for the proof of Theorem 3. Notice that under favorable event $\xi_{t-1}$, we have $U_t(e) - \overline{w}(e) \leq 2c\sqrt{x_e^\intercal \mathbf{M}_{t-1}^{-1} x_e}$ for all $e \in \mathcal{E}$. Therefore, we have

$$\mathbb{E}[R_t^{\alpha\gamma}] \leq \frac{2c}{\alpha\gamma} \mathbb{P}(\xi_{t-1}) \mathbb{E}\left[\sum_{v \in \mathcal{V} \setminus \mathcal{S}_t} \sum_{e \in \mathcal{E}_{\mathcal{S}_t,v}} \mathbf{1}\{O_t(e)\}\sqrt{x_e^\intercal \mathbf{M}_{t-1}^{-1} x_e}\middle|\xi_{t-1}\right] + \mathbb{P}(\overline{\xi}_{t-1}) [L - K]$$

$$\leq \frac{2c}{\alpha\gamma} \mathbb{E}\left[\sum_{v \in \mathcal{V} \setminus \mathcal{S}_t} \sum_{e \in \mathcal{E}_{\mathcal{S}_t,v}} \mathbf{1}\{O_t(e)\}\sqrt{x_e^\intercal \mathbf{M}_{t-1}^{-1} x_e}\right] + \mathbb{P}(\overline{\xi}_{t-1}) [L - K]$$

$$= \frac{2c}{\alpha\gamma} \mathbb{E}\left[\sum_{e \in \mathcal{E}} \mathbf{1}\{O_t(e)\}\sqrt{x_e^\intercal \mathbf{M}_{t-1}^{-1} x_e} \sum_{v \in \mathcal{V} \setminus \mathcal{S}_t} \mathbf{1}\left\{e \in \mathcal{E}_{\mathcal{S}_t,v}\right\}\right] + \mathbb{P}(\overline{\xi}_{t-1}) [L - K]$$

$$= \frac{2c}{\alpha\gamma} \mathbb{E}\left[\sum_{e \in \mathcal{E}} \mathbf{1}\{O_t(e)\} N_{\mathcal{S}_t,e}\sqrt{x_e^\intercal \mathbf{M}_{t-1}^{-1} x_e}\right] + \mathbb{P}(\overline{\xi}_{t-1}) [L - K], \tag{17}$$

where $N_{\mathcal{S}_t,e} = \sum_{v\in\mathcal{V}\setminus\mathcal{S}} \mathbf{1}\left\{e \in \mathcal{E}_{\mathcal{S}_t,v}\right\}$ is defined in Equation 1. Thus we have

$$R^{\alpha\gamma}(n) \le \frac{2c}{\alpha\gamma}\mathbb{E}\left[\sum_{t=1}^{n}\sum_{e\in\mathcal{E}}\mathbf{1}\{O_t(e)\}N_{\mathcal{S}_t,e}\sqrt{x_e^{\intercal}\mathbf{M}_{t-1}^{-1}x_e}\right] + [L-K]\sum_{t=1}^{n}\mathbb{P}\left(\overline{\xi}_{t-1}\right). \qquad (18)$$

In the following lemma, we give a worst-case bound on $\sum_{t=1}^{n}\sum_{e\in\mathcal{E}}\mathbf{1}\{O_t(e)\}N_{\mathcal{S}_t,e}\sqrt{x_e^{\intercal}\mathbf{M}_{t-1}^{-1}x_e}$.

**Lemma 1** *For any round $t = 1, 2, \ldots, n$, we have*

$$\sum_{t=1}^{n}\sum_{e\in\mathcal{E}}\mathbf{1}\{O_t(e)\}N_{\mathcal{S}_t,e}\sqrt{x_e^{\intercal}\mathbf{M}_{t-1}^{-1}x_e} \le \sqrt{\left(\sum_{t=1}^{n}\sum_{e\in\mathcal{E}}\mathbf{1}\{O_t(e)\}N_{\mathcal{S}_t,e}^2\right)\frac{dE_* \log\left(1 + \frac{nE_*}{d\sigma^2}\right)}{\log\left(1 + \frac{1}{\sigma^2}\right)}}.$$

*Moreover, if $X = I \in \Re^{|\mathcal{E}|\times|\mathcal{E}|}$, then we have*

$$\sum_{t=1}^{n}\sum_{e\in\mathcal{E}}\mathbf{1}\{O_t(e)\}N_{\mathcal{S}_t,e}\sqrt{x_e^{\intercal}\mathbf{M}_{t-1}^{-1}x_e} \le \sqrt{\left(\sum_{t=1}^{n}\sum_{e\in\mathcal{E}}\mathbf{1}\{O_t(e)\}N_{\mathcal{S}_t,e}^2\right)\frac{|\mathcal{E}| \log\left(1 + \frac{n}{\sigma^2}\right)}{\log\left(1 + \frac{1}{\sigma^2}\right)}}.$$

Please refer to Section A.3 for the proof of Lemma 1. Finally, notice that for any $t$,

$$\mathbb{E}\left[\sum_{e\in\mathcal{E}}\mathbf{1}\{O_t(e)\}N_{\mathcal{S}_t,e}^2 \,\middle|\, \mathcal{S}_t\right] = \sum_{e\in\mathcal{E}}N_{\mathcal{S}_t,e}^2\mathbb{E}\left[\mathbf{1}\{O_t(e)\}|\mathcal{S}_t\right] = \sum_{e\in\mathcal{E}}N_{\mathcal{S}_t,e}^2 P_{\mathcal{S}_t,e} \le C_*^2,$$

thus taking the expectation over the possibly randomized oracle and Jensen's inequality, we get

$$\mathbb{E}\left[\sqrt{\sum_{t=1}^{n}\sum_{e\in\mathcal{E}}\mathbf{1}\{O_t(e)\}N_{\mathcal{S}_t,e}^2}\right] \le \sqrt{\sum_{t=1}^{n}\mathbb{E}\left[\sum_{e\in\mathcal{E}}\mathbf{1}\{O_t(e)\}N_{\mathcal{S}_t,e}^2\right]} \le \sqrt{\sum_{t=1}^{n}C_*^2} = C_*\sqrt{n}. \quad (19)$$

Combining the above with Lemma 1 and (18), we obtain

$$R^{\alpha\gamma}(n) \le \frac{2cC_*}{\alpha\gamma}\sqrt{\frac{dnE_* \log\left(1 + \frac{nE_*}{d\sigma^2}\right)}{\log\left(1 + \frac{1}{\sigma^2}\right)}} + [L-K]\sum_{t=1}^{n}\mathbb{P}\left(\overline{\xi}_{t-1}\right). \qquad (20)$$

For the special case when $X = I$, we have

$$R^{\alpha\gamma}(n) \le \frac{2cC_*}{\alpha\gamma}\sqrt{\frac{n|\mathcal{E}| \log\left(1 + \frac{n}{\sigma^2}\right)}{\log\left(1 + \frac{1}{\sigma^2}\right)}} + [L-K]\sum_{t=1}^{n}\mathbb{P}\left(\overline{\xi}_{t-1}\right). \qquad (21)$$

Finally, we need to bound the failure probability of upper confidence bound being wrong $\sum_{t=1}^{n}\mathbb{P}\left(\overline{\xi}_{t-1}\right)$. We prove the following bound on $\mathbb{P}\left(\overline{\xi}_{t-1}\right)$:

**Lemma 2** *For any $t = 1, 2, \ldots, n$, any $\sigma > 0$, any $\delta \in (0, 1)$, and any*

$$c \ge \frac{1}{\sigma}\sqrt{d\log\left(1 + \frac{nE_*}{d\sigma^2}\right) + 2\log\left(\frac{1}{\delta}\right)} + \|\theta^*\|_2,$$

*we have $\mathbb{P}\left(\overline{\xi}_{t-1}\right) \le \delta$.*

Please refer to Section A.4 for the proof of Lemma 2. From Lemma 2, for a known upper bound $D$ on $\|\theta^*\|_2$, if we choose $\sigma = 1$ and $c \ge \sqrt{d\log\left(1 + \frac{nE_*}{d}\right) + 2\log\left(n(L+1-K)\right)} + D$, which corresponds to $\delta = \frac{1}{n(L+1-K)}$ in Lemma 2, then we have

$$[L-K]\sum_{t=1}^{n}\mathbb{P}\left(\overline{\xi}_{t-1}\right) < 1.$$

This concludes the proof of Theorem 2. $\qquad\qquad\square$

## A.2    Proof of Theorem 3

Recall that we use $\mathcal{G}_{\mathcal{S}_t,v} = (\mathcal{V}_{\mathcal{S}_t,v}, \mathcal{E}_{\mathcal{S}_t,v})$ to denote the relevant subgraph of node $v$ under the source node set $\mathcal{S}_t$. Since Theorem 3 focuses on the influence from $\mathcal{S}_t$ to $v$, and by definition all the paths from $\mathcal{S}_t$ to $v$ are in $\mathcal{G}_{\mathcal{S}_t,v}$, thus, it is sufficient to restrict to $\mathcal{G}_{\mathcal{S}_t,v}$ and ignore other parts of $\mathcal{G}$ in this analysis.

We start by defining some useful notations.

**Influence Probability with Removed Nodes:** Recall that for any weight function $w : \mathcal{E} \to [0,1]$, any source node set $\mathcal{S} \subset \mathcal{V}$ and any target node $v \in \mathcal{V}$, $f(\mathcal{S}, w, v)$ is the probability that $\mathcal{S}$ will influence $v$ under weight $w$ (see Definition 1). We now define a similar notation for the **influence probability with removed nodes**. Specifically, for any disjoint node set $\mathcal{V}_1, \mathcal{V}_2 \subseteq \mathcal{V}_{\mathcal{S}_t,v} \subseteq \mathcal{V}$, we define $h(\mathcal{V}_1, \mathcal{V}_2, w)$ as follows:

- First, we remove nodes $\mathcal{V}_2$, as well as all edges connected to/from $\mathcal{V}_2$, from $\mathcal{G}_{\mathcal{S}_t,v}$, and obtain a new graph $\mathcal{G}'$.
- $h(\mathcal{V}_1, \mathcal{V}_2, w)$ is the probability that $\mathcal{V}_1$ will influence the target node $v$ in graph $\mathcal{G}'$ under the weight (activation probability) $w(e)$ for all $e \in \mathcal{G}'$.

Obviously, a mathematically equivalent way to define $h(\mathcal{V}_1, \mathcal{V}_2, w)$ is to define it as the probability that $\mathcal{V}_1$ will influence $v$ in $\mathcal{G}_{\mathcal{S}_t,v}$ under a new weight $\widetilde{w}$, defined as

$$\widetilde{w}(e) = \begin{cases} 0 & \text{if } e \text{ is from or to a node in } \mathcal{V}_2 \\ w(e) & \text{otherwise} \end{cases}$$

Note that by definition, $f(\mathcal{S}_t, w, v) = h(\mathcal{S}_t, \emptyset, w)$. Also note that $h(\mathcal{V}_1, \mathcal{V}_2, w)$ implicitly depends on $v$, but we omit $v$ in this notation to simplify the exposition.

**Edge Set $\mathcal{E}(\mathcal{V}_1, \mathcal{V}_2)$:** For any two disjoint node sets $\mathcal{V}_1, \mathcal{V}_2 \subseteq \mathcal{V}_{\mathcal{S}_t,v}$, we define the edge set $\mathcal{E}(\mathcal{V}_1, \mathcal{V}_2)$ as

$$\mathcal{E}(\mathcal{V}_1, \mathcal{V}_2) = \{e = (u_1, u_2) : e \in \mathcal{E}_{\mathcal{S}_t,v}, u_1 \in \mathcal{V}_1, \text{ and } u_2 \notin \mathcal{V}_2\}.$$

That is, $\mathcal{E}(\mathcal{V}_1, \mathcal{V}_2)$ is the set of edges in $\mathcal{G}_{\mathcal{S}_t,v}$ from $\mathcal{V}_1$ to $\mathcal{V}_{\mathcal{S}_t,v} \setminus \mathcal{V}_2$.

**Diffusion Process:** Note that under any edge activation realization $\mathbf{w}(e)$, $e \in \mathcal{E}_{\mathcal{S}_t,v}$, on the relevant subgraph $\mathcal{G}_{\mathcal{S}_t,v}$, we define a finite-length sequence of disjoint node sets $\mathcal{S}^0, \mathcal{S}^1, \ldots, \mathcal{S}^{\widetilde{\tau}}$ as

$$\mathcal{S}^0 \triangleq \mathcal{S}_t$$
$$\mathcal{S}^{\tau+1} \triangleq \left\{ u_2 \in \mathcal{V}_{\mathcal{S}_t,v} : u_2 \notin \cup_{\tau'=0}^{\tau}\mathcal{S}^{\tau'} \text{ and } \exists e = (u_1, u_2) \in \mathcal{E}_{\mathcal{S}_t,v} \text{ s.t. } u_1 \in \mathcal{S}^{\tau} \text{ and } \mathbf{w}(e) = 1 \right\}, \tag{22}$$

$\forall \tau = 0, \ldots, \widetilde{\tau} - 1$. That is, under the realization $\mathbf{w}(e)$, $e \in \mathcal{E}_{\mathcal{S}_t,v}$, $\mathcal{S}^{\tau+1}$ is the set of nodes directly activated by $\mathcal{S}^{\tau}$. Specifically, any node $u_2 \in \mathcal{S}^{\tau+1}$ satisfies $u_2 \notin \bigcup_{\tau'=0}^{\tau} \mathcal{S}^{\tau'}$ (i.e. it was not activated before), and there exists an activated edge $e$ from $\mathcal{S}^{\tau}$ to $u_2$ (i.e. it is activated by some node in $\mathcal{S}^{\tau}$). We define $\mathcal{S}^{\widetilde{\tau}}$ as the first node set in the sequence s.t. either $\mathcal{S}^{\widetilde{\tau}} = \emptyset$ or $v \in \mathcal{S}^{\widetilde{\tau}}$, and assume this sequence terminates at $\mathcal{S}^{\widetilde{\tau}}$. Note that by definition, $\widetilde{\tau} \leq |\mathcal{V}_{\mathcal{S}_t,v}|$ always holds. We refer to each $\tau = 0, 1, \ldots, \widetilde{\tau}$ as a **diffusion step** in this section.

To simplify the exposition, we also define $S^{0:\tau} \triangleq \bigcup_{\tau'=0}^{\tau} S^{\tau'}$ for all $\tau \geq 0$ and $S^{0:-1} \triangleq \emptyset$. Since $\mathbf{w}$ is random, $(\mathcal{S}^{\tau})_{\tau=0}^{\widetilde{\tau}}$ is a stochastic process, which we refer to as the **diffusion process**. Note that $\widetilde{\tau}$ is also random; in particular, it is a stopping time.

Based on the shorthand notations defined above, we have the following lemma for the diffusion process $(\mathcal{S}^{\tau})_{\tau=0}^{\widetilde{\tau}}$ under any weight function $w$:

**Lemma 3** *For any weight function $w : \mathcal{E} \to [0, 1]$, any step $\tau = 0, 1, \ldots, \widetilde{\tau}$, any $\mathcal{S}_\tau$ and $\mathcal{S}^{0:\tau-1}$, we have*

$$h\left(\mathcal{S}^\tau, \mathcal{S}^{0:\tau-1}, w\right) = \begin{cases} 1 & \text{if } v \in \mathcal{S}^\tau \\ 0 & \text{if } \mathcal{S}^\tau = \emptyset \\ \mathbb{E}\left[h\left(\mathcal{S}^{\tau+1}, \mathcal{S}^{0:\tau}, w\right) \big| \left(\mathcal{S}^\tau, \mathcal{S}^{0:\tau-1}\right)\right] & \text{otherwise} \end{cases},$$

*where the expectation is over $\mathcal{S}^{\tau+1}$ under weight $w$. Note that the tuple $(\mathcal{S}^\tau, \mathcal{S}^{0:\tau-1})$ in the conditional expectation means that $\mathcal{S}^\tau$ is the source node set and nodes in $\mathcal{S}^{0:\tau-1}$ have been removed.*

**Proof:** Notice that by definition, $h\left(\mathcal{S}^\tau, \mathcal{S}^{0:\tau-1}, w\right) = 1$ if $v \in \mathcal{S}^\tau$ and $h\left(\mathcal{S}^\tau, \mathcal{S}^{0:\tau-1}, w\right) = 0$ if $\mathcal{S}^\tau = \emptyset$. Also note that in these two cases, $\widetilde{\tau} = \tau$.

Otherwise, we prove that $h\left(\mathcal{S}^\tau, \mathcal{S}^{0:\tau-1}, w\right) = \mathbb{E}\left[h\left(\mathcal{S}^{\tau+1}, \mathcal{S}^{0:\tau}, w\right) \big| \left(\mathcal{S}^\tau, \mathcal{S}^{0:\tau-1}\right)\right]$. Recall that by definition, $h\left(\mathcal{S}^\tau, \mathcal{S}^{0:\tau-1}, w\right)$ is the probability that $v$ will be influenced conditioning on

$$\text{source node set } \mathcal{S}^\tau \text{ and removed node set } \mathcal{S}^{0:\tau-1}, \tag{23}$$

that is

$$h\left(\mathcal{S}^\tau, \mathcal{S}^{0:\tau-1}, w\right) = \mathbb{E}\left[\mathbf{1}\left(v \text{ is influenced}\right) \big| \left(\mathcal{S}^\tau, \mathcal{S}^{0:\tau-1}\right)\right] \tag{24}$$

Let $\mathbf{w}(e), \forall e \in \mathcal{E}(\mathcal{S}^\tau, \mathcal{S}^{0:\tau})$ be any possible realization. Now we analyze the probability that $v$ will be influenced conditioning on

$$\text{source node set } \mathcal{S}^\tau, \text{ removed node set } \mathcal{S}^{0:\tau-1}, \text{ and } \mathbf{w}(e) \text{ for all } e \in \mathcal{E}(\mathcal{S}^\tau, \mathcal{S}^{0:\tau}). \tag{25}$$

Specifically, conditioning on Equation 25, we can define a new weight function $w'$ as

$$w'(e) = \begin{cases} \mathbf{w}(e) & \text{if } e \in \mathcal{E}(\mathcal{S}^\tau, \mathcal{S}^{0:\tau}) \\ w(e) & \text{otherwise} \end{cases} \tag{26}$$

then $h\left(\mathcal{S}^\tau, \mathcal{S}^{0:\tau-1}, w'\right)$ is the probability that $v$ will be influenced conditioning on Equation 25. That is,

$$h\left(\mathcal{S}^\tau, \mathcal{S}^{0:\tau-1}, w'\right) = \mathbb{E}\left[\mathbf{1}\left(v \text{ is influenced}\right) \big| \left(\mathcal{S}^\tau, \mathcal{S}^{0:\tau-1}\right), \mathbf{w}(e) \,\forall e \in \mathcal{E}(\mathcal{S}^\tau, \mathcal{S}^{0:\tau})\right], \tag{27}$$

for any possible realization of $\mathbf{w}(e), \forall e \in \mathcal{E}(\mathcal{S}^\tau, \mathcal{S}^{0:\tau})$. Notice that on the lefthand of Equation 27, $w'$ encodes the conditioning on $\mathbf{w}(e)$ for all $e \in \mathcal{E}(\mathcal{S}^\tau, \mathcal{S}^{0:\tau})$ (see Equation 26).

From here to Equation 29, we focus on an arbitrary but fixed realization of $\mathbf{w}(e), \forall e \in \mathcal{E}(\mathcal{S}^\tau, \mathcal{S}^{0:\tau})$ (or equivalently, an arbitrary but fixed $w'$). Based on the definition of $\mathcal{S}^{\tau+1}$, conditioning on Equation 25, $\mathcal{S}^{\tau+1}$ is deterministic and all nodes in $\mathcal{S}^{\tau+1}$ can also be treated as source nodes. Thus, we have

$$h\left(\mathcal{S}^\tau, \mathcal{S}^{0:\tau-1}, w'\right) = h\left(\mathcal{S}^\tau \cup \mathcal{S}^{\tau+1}, \mathcal{S}^{0:\tau-1}, w'\right),$$

conditioning on Equation 25.

On the other hand, conditioning on Equation 25, we can treat any edge $e \in \mathcal{E}(\mathcal{S}^\tau, \mathcal{S}^{0:\tau})$ with $\mathbf{w}(e) = 0$ as having been removed. Since nodes in $\mathcal{S}^{0:\tau-1}$ have also been removed, and $v \notin \mathcal{S}^\tau$, then if there is a path from $\mathcal{S}^\tau$ to $v$, then it must go through $\mathcal{S}^{\tau+1}$, and the last node on the path in $\mathcal{S}^{\tau+1}$ must be after the last node on the path in $\mathcal{S}^\tau$ (note that the path might come back to $\mathcal{S}^\tau$ for several times). Hence, conditioning on Equation 25, if nodes in $\mathcal{S}^{\tau+1}$ are also treated as source nodes, then $\mathcal{S}^\tau$ is irrelevant for influence on $v$ and can be removed. So we have

$$h\left(\mathcal{S}^\tau, \mathcal{S}^{0:\tau-1}, w'\right) = h\left(\mathcal{S}^\tau \cup \mathcal{S}^{\tau+1}, \mathcal{S}^{0:\tau-1}, w'\right) = h\left(\mathcal{S}^{\tau+1}, \mathcal{S}^{0:\tau}, w\right). \tag{28}$$

Note that in the last equation we change the weight function back to $w$ since edges in $\mathcal{E}(\mathcal{S}^\tau, \mathcal{S}^{0:\tau})$ have been removed. Thus, conditioning on Equation 25, we have

$$\begin{aligned} h\left(\mathcal{S}^{\tau+1}, \mathcal{S}^{0:\tau}, w\right) &= h\left(\mathcal{S}^\tau, \mathcal{S}^{0:\tau-1}, w'\right) \\ &= \mathbb{E}\left[\mathbf{1}\left(v \text{ is influenced}\right) \big| \left(\mathcal{S}^\tau, \mathcal{S}^{0:\tau-1}\right), \mathbf{w}(e) \,\forall e \in \mathcal{E}(\mathcal{S}^\tau, \mathcal{S}^{0:\tau})\right]. \end{aligned} \tag{29}$$

Notice again that Equation 29 holds for any possible realization of $\mathbf{w}(e), \forall e \in \mathcal{E}(\mathcal{S}^\tau, \mathcal{S}^{0:\tau})$.

Finally, we have

$$h\left(\mathcal{S}^\tau, \mathcal{S}^{0:\tau-1}, w\right) \overset{(a)}{=} \mathbb{E}\left[\mathbf{1}\left(v \text{ is influenced}\right)\middle|\left(\mathcal{S}^\tau, \mathcal{S}^{0:\tau-1}\right)\right]$$

$$\overset{(b)}{=} \mathbb{E}\left[\mathbb{E}\left[\mathbf{1}\left(v \text{ is influenced}\right)\middle|\left(\mathcal{S}^\tau, \mathcal{S}^{0:\tau-1}\right), \mathbf{w}(e)\,\forall e \in \mathcal{E}(\mathcal{S}^\tau, \mathcal{S}^{0:\tau})\right]\middle|\left(\mathcal{S}^\tau, \mathcal{S}^{0:\tau-1}\right)\right]$$

$$\overset{(c)}{=} \mathbb{E}\left[h\left(\mathcal{S}^{\tau+1}, \mathcal{S}^{0:\tau}, w\right)\middle|\left(\mathcal{S}^\tau, \mathcal{S}^{0:\tau-1}\right)\right], \tag{30}$$

where (a) follows from Equation 24, (b) follows from the tower rule, and (c) follows from Equation 29. This concludes the proof. $\qquad\square$

Consider two weight functions $U, w : \mathcal{E} \to [0,1]$ s.t. $U(e) \geq w(e)$ for all $e \in \mathcal{E}$. The following lemma bounds the difference $h\left(\mathcal{S}^\tau, \mathcal{S}^{0:\tau-1}, U\right) - h\left(\mathcal{S}^\tau, \mathcal{S}^{0:\tau-1}, w\right)$ in a recursive way.

**Lemma 4** *For any two weight functions $w, U : \mathcal{E} \to [0,1]$ s.t. $U(e) \geq w(e)$ for all $e \in \mathcal{E}$, any step $\tau = 0, 1, \ldots, \widetilde{\tau}$, any $\mathcal{S}_\tau$ and $\mathcal{S}^{0:\tau-1}$, we have*

$$h\left(\mathcal{S}^\tau, \mathcal{S}^{0:\tau-1}, U\right) - h\left(\mathcal{S}^\tau, \mathcal{S}^{0:\tau-1}, w\right) = 0$$

*if $v \in \mathcal{S}^\tau$ or $\mathcal{S}^\tau = \emptyset$; and otherwise*

$$h\left(\mathcal{S}^\tau, \mathcal{S}^{0:\tau-1}, U\right) - h\left(\mathcal{S}^\tau, \mathcal{S}^{0:\tau-1}, w\right) \leq \sum_{e \in \mathcal{E}(\mathcal{S}^\tau, \mathcal{S}^{0:\tau})} [U(e) - w(e)]$$

$$+ \mathbb{E}\left[h\left(\mathcal{S}^{\tau+1}, \mathcal{S}^{0:\tau}, U\right) - h\left(\mathcal{S}^{\tau+1}, \mathcal{S}^{0:\tau}, w\right)\middle|\left(\mathcal{S}^\tau, \mathcal{S}^{0:\tau-1}\right)\right],$$

*where the expectation is over $\mathcal{S}^{\tau+1}$ under weight $w$. Recall that the tuple $\left(\mathcal{S}^\tau, \mathcal{S}^{0:\tau-1}\right)$ in the conditional expectation means that $\mathcal{S}^\tau$ is the source node set and nodes in $\mathcal{S}^{0:\tau-1}$ have been removed.*

**Proof:**  First, note that if $v \in \mathcal{S}^\tau$ or $\mathcal{S}^\tau = \emptyset$, then

$$h\left(\mathcal{S}^\tau, \mathcal{S}^{0:\tau-1}, U\right) - h\left(\mathcal{S}^\tau, \mathcal{S}^{0:\tau-1}, w\right) = 0$$

follows directly from Lemma 3. Otherwise, to simplify the exposition, we overload the notation and use $w(\mathcal{S}^{\tau+1})$ to denote the conditional probability of $\mathcal{S}^{\tau+1}$ conditioning on $(\mathcal{S}^\tau, \mathcal{S}^{0:\tau-1})$ under the weight function $w$, and similarly for $U(\mathcal{S}^{\tau+1})$. That is

$$w(\mathcal{S}^{\tau+1}) \overset{\Delta}{=} \text{Prob}\left[\mathcal{S}^{\tau+1}\middle|(\mathcal{S}^\tau, \mathcal{S}^{0:\tau-1}); w\right]$$

$$U(\mathcal{S}^{\tau+1}) \overset{\Delta}{=} \text{Prob}\left[\mathcal{S}^{\tau+1}\middle|(\mathcal{S}^\tau, \mathcal{S}^{0:\tau-1}); U\right], \tag{31}$$

where the tuple $(\mathcal{S}^\tau, \mathcal{S}^{0:\tau-1})$ in the conditional probability means that $\mathcal{S}^\tau$ is the source node set and nodes in $\mathcal{S}^{0:\tau-1}$ have been removed, and $w$ and $U$ after the semicolon indicate the weight function.

Then from Lemma 3, we have

$$h\left(\mathcal{S}^\tau, \mathcal{S}^{0:\tau-1}, U\right) = \sum_{\mathcal{S}^{\tau+1}} U(\mathcal{S}^{\tau+1})h\left(\mathcal{S}^{\tau+1}, \mathcal{S}^{0:\tau}, U\right)$$

$$h\left(\mathcal{S}^\tau, \mathcal{S}^{0:\tau-1}, w\right) = \sum_{\mathcal{S}^{\tau+1}} w(\mathcal{S}^{\tau+1})h\left(\mathcal{S}^{\tau+1}, \mathcal{S}^{0:\tau}, w\right)$$

where the sum is over all possible realization of $\mathcal{S}^{\tau+1}$.

Hence we have

$$h\left(\mathcal{S}^\tau, \mathcal{S}^{0:\tau-1}, U\right) - h\left(\mathcal{S}^\tau, \mathcal{S}^{0:\tau-1}, w\right)$$

$$= \sum_{\mathcal{S}^{\tau+1}} \left[ U(\mathcal{S}^{\tau+1}) h\left(\mathcal{S}^{\tau+1}, \mathcal{S}^{0:\tau}, U\right) - w(\mathcal{S}^{\tau+1}) h\left(\mathcal{S}^{\tau+1}, \mathcal{S}^{0:\tau}, w\right) \right]$$

$$= \sum_{\mathcal{S}^{\tau+1}} \left[ U(\mathcal{S}^{\tau+1}) h\left(\mathcal{S}^{\tau+1}, \mathcal{S}^{0:\tau}, U\right) - w(\mathcal{S}^{\tau+1}) h\left(\mathcal{S}^{\tau+1}, \mathcal{S}^{0:\tau}, U\right) \right]$$

$$+ \sum_{\mathcal{S}^{\tau+1}} \left[ w(\mathcal{S}^{\tau+1}) h\left(\mathcal{S}^{\tau+1}, \mathcal{S}^{0:\tau}, U\right) - w(\mathcal{S}^{\tau+1}) h\left(\mathcal{S}^{\tau+1}, \mathcal{S}^{0:\tau}, w\right) \right]$$

$$= \sum_{\mathcal{S}^{\tau+1}} \left[ U(\mathcal{S}^{\tau+1}) - w(\mathcal{S}^{\tau+1}) \right] h\left(\mathcal{S}^{\tau+1}, \mathcal{S}^{0:\tau}, U\right)$$

$$+ \sum_{\mathcal{S}^{\tau+1}} w(\mathcal{S}^{\tau+1}) \left[ h\left(\mathcal{S}^{\tau+1}, \mathcal{S}^{0:\tau}, U\right) - h\left(\mathcal{S}^{\tau+1}, \mathcal{S}^{0:\tau}, w\right) \right], \tag{32}$$

where the sum in the above equations is also over all the possible realizations of $\mathcal{S}^{\tau+1}$. Notice that by definition, we have

$$\mathbb{E}\left[ h\left(\mathcal{S}^{\tau+1}, \mathcal{S}^{0:\tau}, U\right) - h\left(\mathcal{S}^{\tau+1}, \mathcal{S}^{0:\tau}, w\right) \middle| (\mathcal{S}^\tau, \mathcal{S}^{0:\tau-1}) \right] =$$

$$\sum_{\mathcal{S}^{\tau+1}} w(\mathcal{S}^{\tau+1}) \left[ h\left(\mathcal{S}^{\tau+1}, \mathcal{S}^{0:\tau}, U\right) - h\left(\mathcal{S}^{\tau+1}, \mathcal{S}^{0:\tau}, w\right) \right], \tag{33}$$

where the expectation in the lefthand side is over $\mathcal{S}^{\tau+1}$ under weight $w$, or equivalently, over $\mathbf{w}(e)$ for all $e \in \mathcal{E}(\mathcal{S}^\tau, \mathcal{S}^{0:\tau})$ under weight $w$. Thus, to prove Lemma 4, it is sufficient to prove that

$$\sum_{\mathcal{S}^{\tau+1}} \left[ U(\mathcal{S}^{\tau+1}) - w(\mathcal{S}^{\tau+1}) \right] h\left(\mathcal{S}^{\tau+1}, \mathcal{S}^{0:\tau}, U\right) \leq \sum_{e \in \mathcal{E}(\mathcal{S}^\tau, \mathcal{S}^{0:\tau})} \left[ U(e) - w(e) \right]. \tag{34}$$

Notice that

$$\sum_{\mathcal{S}^{\tau+1}} \left[ U(\mathcal{S}^{\tau+1}) - w(\mathcal{S}^{\tau+1}) \right] h\left(\mathcal{S}^{\tau+1}, \mathcal{S}^{0:\tau}, U\right)$$

$$\overset{(a)}{\leq} \sum_{\mathcal{S}^{\tau+1}} \left[ U(\mathcal{S}^{\tau+1}) - w(\mathcal{S}^{\tau+1}) \right] h\left(\mathcal{S}^{\tau+1}, \mathcal{S}^{0:\tau}, U\right) \mathbf{1}\left[ U(\mathcal{S}^{\tau+1}) \geq w(\mathcal{S}^{\tau+1}) \right]$$

$$\overset{(b)}{\leq} \sum_{\mathcal{S}^{\tau+1}} \left[ U(\mathcal{S}^{\tau+1}) - w(\mathcal{S}^{\tau+1}) \right] \mathbf{1}\left[ U(\mathcal{S}^{\tau+1}) \geq w(\mathcal{S}^{\tau+1}) \right]$$

$$\overset{(c)}{=} \frac{1}{2} \sum_{\mathcal{S}^{\tau+1}} \left| U(\mathcal{S}^{\tau+1}) - w(\mathcal{S}^{\tau+1}) \right|, \tag{35}$$

where (a) holds since

$$\sum_{\mathcal{S}^{\tau+1}} \left[ U(\mathcal{S}^{\tau+1}) - w(\mathcal{S}^{\tau+1}) \right] h\left(\mathcal{S}^{\tau+1}, \mathcal{S}^{0:\tau}, U\right) =$$

$$\sum_{\mathcal{S}^{\tau+1}} \left[ U(\mathcal{S}^{\tau+1}) - w(\mathcal{S}^{\tau+1}) \right] h\left(\mathcal{S}^{\tau+1}, \mathcal{S}^{0:\tau}, U\right) \mathbf{1}\left[ U(\mathcal{S}^{\tau+1}) \geq w(\mathcal{S}^{\tau+1}) \right]$$

$$+ \sum_{\mathcal{S}^{\tau+1}} \left[ U(\mathcal{S}^{\tau+1}) - w(\mathcal{S}^{\tau+1}) \right] h\left(\mathcal{S}^{\tau+1}, \mathcal{S}^{0:\tau}, U\right) \mathbf{1}\left[ U(\mathcal{S}^{\tau+1}) < w(\mathcal{S}^{\tau+1}) \right],$$

and the second term on the righthand side is non-positive. And (b) holds since $0 \leq h\left(\mathcal{S}^{\tau+1}, \mathcal{S}^{0:\tau}, U\right) \leq 1$ by definition. To prove (c), we define shorthand notations

$$A^+ = \sum_{\mathcal{S}^{\tau+1}} \left[U(\mathcal{S}^{\tau+1}) - w(\mathcal{S}^{\tau+1})\right] \mathbf{1}\left[U(\mathcal{S}^{\tau+1}) \geq w(\mathcal{S}^{\tau+1})\right]$$

$$A^- = \sum_{\mathcal{S}^{\tau+1}} \left[U(\mathcal{S}^{\tau+1}) - w(\mathcal{S}^{\tau+1})\right] \mathbf{1}\left[U(\mathcal{S}^{\tau+1}) < w(\mathcal{S}^{\tau+1})\right]$$

Then we have

$$A^+ + A^- = \sum_{\mathcal{S}^{\tau+1}} \left[U(\mathcal{S}^{\tau+1}) - w(\mathcal{S}^{\tau+1})\right] = 0,$$

since by definition $\sum_{\mathcal{S}^{\tau+1}} U(\mathcal{S}^{\tau+1}) = \sum_{\mathcal{S}^{\tau+1}} w(\mathcal{S}^{\tau+1}) = 1$. Moreover, we also have

$$A^+ - A^- = \sum_{\mathcal{S}^{\tau+1}} \left|U(\mathcal{S}^{\tau+1}) - w(\mathcal{S}^{\tau+1})\right|.$$

And hence $A^+ = \frac{1}{2} \sum_{\mathcal{S}^{\tau+1}} \left|U(\mathcal{S}^{\tau+1}) - w(\mathcal{S}^{\tau+1})\right|$. Thus, to prove Lemma 4, it is sufficient to prove

$$\frac{1}{2} \sum_{\mathcal{S}^{\tau+1}} \left|U(\mathcal{S}^{\tau+1}) - w(\mathcal{S}^{\tau+1})\right| \leq \sum_{e \in \mathcal{E}(\mathcal{S}^\tau, \mathcal{S}^{0:\tau})} \left[U(e) - w(e)\right]. \tag{36}$$

Let $\widetilde{\mathbf{w}} \in \{0,1\}^{|\mathcal{E}(\mathcal{S}^\tau, \mathcal{S}^{0:\tau})|}$ be an arbitrary edge activation realization for edges in $\mathcal{E}(\mathcal{S}^\tau, \mathcal{S}^{0:\tau})$. Also with a little bit abuse of notation, we use $w(\widetilde{\mathbf{w}})$ to denote the probability of $\widetilde{\mathbf{w}}$ under weight $w$. Notice that

$$w(\widetilde{\mathbf{w}}) = \prod_{e \in \mathcal{E}(\mathcal{S}^\tau, \mathcal{S}^{0:\tau})} w(e)^{\widetilde{\mathbf{w}}(e)} \left[1 - w(e)\right]^{1-\widetilde{\mathbf{w}}(e)},$$

and $U(\widetilde{\mathbf{w}})$ is defined similarly. Recall that by definition $\mathcal{S}^{\tau+1}$ is a deterministic function of source node set $\mathcal{S}^\tau$, removed nodes $\mathcal{S}^{0:\tau-1}$, and $\widetilde{\mathbf{w}}$. Hence, for any possible realized $\mathcal{S}^{\tau+1}$, let $\mathbf{W}(\mathcal{S}^{\tau+1})$ denote the set of $\widetilde{\mathbf{w}}$'s that lead to this $\mathcal{S}^{\tau+1}$, then we have

$$U(\mathcal{S}^{\tau+1}) = \sum_{\widetilde{\mathbf{w}} \in \mathbf{W}(\mathcal{S}^{\tau+1})} U(\widetilde{\mathbf{w}}) \quad \text{and} \quad w(\mathcal{S}^{\tau+1}) = \sum_{\widetilde{\mathbf{w}} \in \mathbf{W}(\mathcal{S}^{\tau+1})} w(\widetilde{\mathbf{w}})$$

Thus, we have

$$\begin{aligned}
\frac{1}{2} \sum_{\mathcal{S}^{\tau+1}} \left|U(\mathcal{S}^{\tau+1}) - w(\mathcal{S}^{\tau+1})\right| &= \frac{1}{2} \sum_{\mathcal{S}^{\tau+1}} \left|\sum_{\widetilde{\mathbf{w}} \in \mathbf{W}(\mathcal{S}^{\tau+1})} \left[U(\widetilde{\mathbf{w}}) - w(\widetilde{\mathbf{w}})\right]\right| \\
&\leq \frac{1}{2} \sum_{\mathcal{S}^{\tau+1}} \sum_{\widetilde{\mathbf{w}} \in \mathbf{W}(\mathcal{S}^{\tau+1})} \left|U(\widetilde{\mathbf{w}}) - w(\widetilde{\mathbf{w}})\right| \\
&= \frac{1}{2} \sum_{\widetilde{\mathbf{w}}} \left|U(\widetilde{\mathbf{w}}) - w(\widetilde{\mathbf{w}})\right| \tag{37}
\end{aligned}$$

Finally, we prove that

$$\frac{1}{2} \sum_{\widetilde{\mathbf{w}}} \left|U(\widetilde{\mathbf{w}}) - w(\widetilde{\mathbf{w}})\right| \leq \sum_{e \in \mathcal{E}(\mathcal{S}^\tau, \mathcal{S}^{0:\tau})} \left[U(e) - w(e)\right] \tag{38}$$

by mathematical induction. Without loss of generality, we order the edges in $\mathcal{E}(\mathcal{S}^\tau, \mathcal{S}^{0:\tau})$ as $1, 2, \ldots, |\mathcal{E}(\mathcal{S}^\tau, \mathcal{S}^{0:\tau})|$. For any $k = 1, \ldots, |\mathcal{E}(\mathcal{S}^\tau, \mathcal{S}^{0:\tau})|$, we use $\widetilde{\mathbf{w}}_k \in \{0,1\}^k$ to denote an arbitrary edge activation realization for edges $1, \ldots, k$. Then, we prove

$$\frac{1}{2} \sum_{\widetilde{\mathbf{w}}_k} \left|U(\widetilde{\mathbf{w}}_k) - w(\widetilde{\mathbf{w}}_k)\right| \leq \sum_{e=1}^{k} \left[U(e) - w(e)\right] \tag{39}$$

for all $k = 1, \ldots, |\mathcal{E}(\mathcal{S}^\tau, \mathcal{S}^{0:\tau})|$ by mathematical induction. Notice that when $k = 1$, we have

$$\frac{1}{2} \sum_{\widetilde{\mathbf{w}}_1} |U(\widetilde{\mathbf{w}}_1) - w(\widetilde{\mathbf{w}}_1)| = \frac{1}{2} \left[ |U(1) - w(1)| + |(1 - U(1)) - (1 - w(1))| \right] = U(1) - w(1).$$

Now assume that the induction hypothesis holds for $k$, we prove that it also holds for $k + 1$. Note that

$$
\begin{aligned}
\frac{1}{2} \sum_{\widetilde{\mathbf{w}}_{k+1}} |U(\widetilde{\mathbf{w}}_{k+1}) - w(\widetilde{\mathbf{w}}_{k+1})| =& \frac{1}{2} \sum_{\widetilde{\mathbf{w}}_k} \big[ |U(\widetilde{\mathbf{w}}_k) U(k+1) - w(\widetilde{\mathbf{w}}_k) w(k+1)| \\
& + |U(\widetilde{\mathbf{w}}_k)(1 - U(k+1)) - w(\widetilde{\mathbf{w}}_k)(1 - w(k+1))| \big] \\
\overset{(a)}{\leq}& \frac{1}{2} \sum_{\widetilde{\mathbf{w}}_k} \big[ |U(\widetilde{\mathbf{w}}_k) U(k+1) - w(\widetilde{\mathbf{w}}_k) U(k+1)| \\
& + |w(\widetilde{\mathbf{w}}_k) U(k+1) - w(\widetilde{\mathbf{w}}_k) w(k+1)| \\
& + |U(\widetilde{\mathbf{w}}_k)(1 - U(k+1)) - w(\widetilde{\mathbf{w}}_k)(1 - U(k+1))| \\
& + |w(\widetilde{\mathbf{w}}_k)(1 - U(k+1)) - w(\widetilde{\mathbf{w}}_k)(1 - w(k+1))| \big] \\
=& \frac{1}{2} \sum_{\widetilde{\mathbf{w}}_k} \big[ U(k+1) |U(\widetilde{\mathbf{w}}_k) - w(\widetilde{\mathbf{w}}_k)| + w(\widetilde{\mathbf{w}}_k) |U(k+1) - w(k+1)| \\
& + (1 - U(k+1)) |U(\widetilde{\mathbf{w}}_k) - w(\widetilde{\mathbf{w}}_k)| + w(\widetilde{\mathbf{w}}_k) |U(k+1) - w(k+1)| \big] \\
=& \frac{1}{2} \sum_{\widetilde{\mathbf{w}}_k} |U(\widetilde{\mathbf{w}}_k) - w(\widetilde{\mathbf{w}}_k)| + [U(k+1) - w(k+1)] \\
\overset{(b)}{\leq}& \sum_{e=1}^{k} [U(e) - w(e)] + [U(k+1) - w(k+1)] \\
=& \sum_{e=1}^{k+1} [U(e) - w(e)], \quad\quad\quad\quad\quad\quad\quad\quad\quad\quad\quad (40)
\end{aligned}
$$

where (a) follows from the triangular inequality and (b) follows from the induction hypothesis. Hence, we have proved Equation 39 by induction hypothesis. As we have proved above, this is sufficient to prove Lemma 4. $\qquad\square$

Finally, we prove the following lemma:

**Lemma 5** *For any two weight functions $w, U : \mathcal{E} \to [0, 1]$ s.t. $U(e) \geq w(e)$ for all $e \in \mathcal{E}$, we have*

$$f(\mathcal{S}_t, U, v) - f(\mathcal{S}_t, w, v) \leq \mathbb{E}\left[ \sum_{\tau=0}^{\widetilde{\tau}-1} \sum_{e \in \mathcal{E}(\mathcal{S}^\tau, \mathcal{S}^{0:\tau})} [U(e) - w(e)] \Big| \mathcal{S}_t \right],$$

*where $\widetilde{\tau}$ is the stopping time when $\mathcal{S}^\tau = \emptyset$ or $v \in \mathcal{S}^\tau$, and the expectation is under the weight function $w$.*

**Proof:** Recall that the diffusion process $(\mathcal{S}^\tau)_{\tau=0}^{\widetilde{\tau}}$ is a stochastic process. Note that by definition, if we treat the pair $(\mathcal{S}^\tau, \mathcal{S}^{0:\tau-1})$ as the *state* of the diffusion process at diffusion step $\tau$, and assume that $\mathbf{w}(e) \sim \text{Bern}(w(e))$ are independently sampled for all $e \in \mathcal{E}_{\mathcal{S}_t, v}$, then the sequence $(\mathcal{S}^0, \mathcal{S}^{0:-1}), (\mathcal{S}^0, \mathcal{S}^{0:-1}), \ldots, (\mathcal{S}^{\widetilde{\tau}}, \mathcal{S}^{0:\widetilde{\tau}-1})$ follows a Markov chain, specifically,

- For any state $(\mathcal{S}^\tau, \mathcal{S}^{0:\tau-1})$ s.t. $v \notin \mathcal{S}^\tau$ and $\mathcal{S}^\tau \neq \emptyset$, its transition probabilities to the next state $(\mathcal{S}^{\tau+1}, \mathcal{S}^{0:\tau})$ depend on $w(e)$'s for $e \in \mathcal{E}(\mathcal{S}^\tau, \mathcal{S}^{0:\tau})$.

- Any state $(\mathcal{S}^\tau, \mathcal{S}^{0:\tau-1})$ s.t. $v \in \mathcal{S}^\tau$ or $\mathcal{S}^\tau = \emptyset$ is a terminal state and the state transition terminates once visiting such a state. Recall that by definition of the stopping time $\widetilde{\tau}$, the state transition terminates at $\widetilde{\tau}$.

We define $h(\mathcal{S}^\tau, \mathcal{S}^{0:\tau-1}, U) - h(\mathcal{S}^\tau, \mathcal{S}^{0:\tau-1}, w)$ as the "value" at state $(\mathcal{S}^\tau, \mathcal{S}^{0:\tau-1})$. Also note that the states in this Markov chain is *topologically sortable* in the sense that it will never revisit a state it visits before. Hence, we can compute $h(\mathcal{S}^\tau, \mathcal{S}^{0:\tau-1}, U) - h(\mathcal{S}^\tau, \mathcal{S}^{0:\tau-1}, w)$ via a backward

induction from the terminal states, based on a valid topological order. Thus, from Lemma 4, we have

$$f(\mathcal{S}_t, U, v) - f(\mathcal{S}_t, w, v) \overset{(a)}{=} h(\mathcal{S}^0, \emptyset, U) - h(\mathcal{S}^0, \emptyset, w)$$

$$\overset{(b)}{\leq} \mathbb{E}\left[\sum_{\tau=0}^{\widetilde{\tau}-1} \sum_{e \in \mathcal{E}(\mathcal{S}^\tau, \mathcal{S}^{0:\tau})} [U(e) - w(e)] \Big| \mathcal{S}^0\right], \tag{41}$$

where $(a)$ follows from the definition of $h$, and (b) follows from the backward induction. Since $\mathcal{S}^0 = \mathcal{S}_t$ by definition, we have proved Lemma 5. $\qquad\square$

Finally, we prove Theorem 3 based on Lemma 5. Recall that the favorable event at round $t-1$ is defined as

$$\xi_{t-1} = \left\{ |x_e^\intercal(\overline{\theta}_{\tau-1} - \theta^*)| \leq c\sqrt{x_e^\intercal \mathbf{M}_{\tau-1}^{-1} x_e}, \forall e \in \mathcal{E}, \forall \tau \leq t \right\}.$$

Also, based on Algorithm 1, we have

$$0 \leq \overline{w}(e) \leq U_t(e) \leq 1, \forall e \in \mathcal{E}.$$

Thus, from Lemma 5, we have

$$f(\mathcal{S}_t, U_t, v) - f(\mathcal{S}_t, \overline{w}, v) \leq \mathbb{E}\left[\sum_{\tau=0}^{\widetilde{\tau}-1} \sum_{e \in \mathcal{E}(\mathcal{S}^\tau, \mathcal{S}^{0:\tau})} [U_t(e) - \overline{w}(e)] \Big| \mathcal{S}_t, \mathcal{H}_{t-1}\right],$$

where the expectation is based on the weight function $\overline{w}$. Recall that $O_t(e)$ is the event that edge $e$ is observed at round $t$. Recall that by definition, all edges in $\mathcal{E}(\mathcal{S}^\tau, \mathcal{S}^{0:\tau})$ are observed at round $t$ (since they are going out from an influenced node in $\mathcal{S}^\tau$, see Definition 2) and belong to $\mathcal{E}_{\mathcal{S}_t, v}$, so we have

$$f(\mathcal{S}_t, U_t, v) - f(\mathcal{S}_t, \overline{w}, v) \leq \mathbb{E}\left[\sum_{\tau=0}^{\widetilde{\tau}-1} \sum_{e \in \mathcal{E}(\mathcal{S}^\tau, \mathcal{S}^{0:\tau})} [U_t(e) - \overline{w}(e)] \Big| \mathcal{S}_t, \mathcal{H}_{t-1}\right]$$

$$\leq \mathbb{E}\left[\sum_{e \in \mathcal{E}_{\mathcal{S}_t, v}} \mathbf{1}\left(O_t(e)\right)[U_t(e) - \overline{w}(e)] \Big| \mathcal{S}_t, \mathcal{H}_{t-1}\right]. \tag{42}$$

This completes the proof for Theorem 3.

## A.3    Proof of Lemma 1

**Proof:**    To simplify the exposition, we define $z_{t,e} = \sqrt{x_e^\intercal \mathbf{M}_{t-1}^{-1} x_e}$ for all $t = 1, 2 \ldots, n$ and all $e \in \mathcal{E}$, and use $\mathcal{E}_t^o$ denote the set of edges observed at round $t$. Recall that

$$\mathbf{M}_t = \mathbf{M}_{t-1} + \frac{1}{\sigma^2}\sum_{e \in \mathcal{E}} x_e x_e^\intercal \mathbf{1}\left\{O_t(e)\right\} = \mathbf{M}_{t-1} + \frac{1}{\sigma^2}\sum_{e \in \mathcal{E}_t^o} x_e x_e^\intercal. \tag{43}$$

Thus, for all $(t, e)$ such that $e \in \mathcal{E}_t^o$ (i.e., edge $e$ is observed at round $t$), we have that

$$\det[\mathbf{M}_t] \geq \det\left[\mathbf{M}_{t-1} + \frac{1}{\sigma^2}x_e x_e^\intercal\right] = \det\left[\mathbf{M}_{t-1}^{\frac{1}{2}}\left(\mathbf{I} + \frac{1}{\sigma^2}\mathbf{M}_{t-1}^{-\frac{1}{2}} x_e x_e^\intercal \mathbf{M}_{t-1}^{-\frac{1}{2}}\right)\mathbf{M}_{t-1}^{\frac{1}{2}}\right]$$

$$= \det[\mathbf{M}_{t-1}]\det\left[\mathbf{I} + \frac{1}{\sigma^2}\mathbf{M}_{t-1}^{-\frac{1}{2}} x_e x_e^\intercal \mathbf{M}_{t-1}^{-\frac{1}{2}}\right]$$

$$= \det[\mathbf{M}_{t-1}]\left(1 + \frac{1}{\sigma^2}x_e^\intercal \mathbf{M}_{t-1}^{-1} x_e\right) = \det[\mathbf{M}_{t-1}]\left(1 + \frac{z_{t,e}^2}{\sigma^2}\right).$$

Thus, we have

$$(\det[\mathbf{M}_t])^{|\mathcal{E}_t^o|} \geq (\det[\mathbf{M}_{t-1}])^{|\mathcal{E}_t^o|}\prod_{e \in \mathcal{E}_t^o}\left(1 + \frac{z_{t,e}^2}{\sigma^2}\right).$$

**Remark 1** *Notice that when the feature matrix $\mathbf{X} = \mathbf{I}$, $\mathbf{M}_t$'s are always diagonal matrices, and we have*

$$\det[\mathbf{M}_t] = \det[\mathbf{M}_{t-1}] \prod_{e \in \mathcal{E}_t^o} \left(1 + \frac{z_{t,e}^2}{\sigma^2}\right),$$

*which will lead to a tighter bound in the tabular ($\mathbf{X} = \mathbf{I}$) case.*

Since 1) $\det[\mathbf{M}_t] \geq \det[\mathbf{M}_{t-1}]$ from Equation 43 and 2) $|\mathcal{E}_t^o| \leq E_*$, where $E_*$ is defined in Equation 10 and $|\mathcal{E}_t^o| \leq E_*$ follows from its definition, we have

$$(\det[\mathbf{M}_t])^{E_*} \geq (\det[\mathbf{M}_{t-1}])^{E_*} \prod_{e \in \mathcal{E}_t^o} \left(1 + \frac{z_{t,e}^2}{\sigma^2}\right).$$

Therefore, we have

$$(\det[\mathbf{M}_n])^{E_*} \geq (\det[\mathbf{M}_0])^{E_*} \prod_{t=1}^{n} \prod_{e \in \mathcal{E}_t^o} \left(1 + \frac{z_{t,e}^2}{\sigma^2}\right) = \prod_{t=1}^{n} \prod_{e \in \mathcal{E}_t^o} \left(1 + \frac{z_{t,e}^2}{\sigma^2}\right),$$

since $\mathbf{M}_0 = \mathbf{I}$. On the other hand, we have that

$$\text{trace}(\mathbf{M}_n) = \text{trace}\left(\mathbf{I} + \frac{1}{\sigma^2} \sum_{t=1}^{n} \sum_{e \in \mathcal{E}_t^o} x_e x_e^\mathsf{T}\right) = d + \frac{1}{\sigma^2} \sum_{t=1}^{n} \sum_{e \in \mathcal{E}_t^o} \|x_e\|_2^2 \leq d + \frac{nE_*}{\sigma^2},$$

where the last inequality follows from the fact that $\|x_e\|_2 \leq 1$ and $|\mathcal{E}_t^o| \leq E_*$. From the trace-determinant inequality, we have $\frac{1}{d}\text{trace}(\mathbf{M}_n) \geq [\det(\mathbf{M}_n)]^{\frac{1}{d}}$, thus we have

$$\left[1 + \frac{nE_*}{d\sigma^2}\right]^{dE_*} \geq \left[\frac{1}{d}\text{trace}(\mathbf{M}_n)\right]^{dE_*} \geq [\det(\mathbf{M}_n)]^{E_*} \geq \prod_{t=1}^{n} \prod_{e \in \mathcal{E}_t^o} \left(1 + \frac{z_{t,e}^2}{\sigma^2}\right).$$

Taking the logarithm on the both sides, we have

$$dE_* \log\left[1 + \frac{nE_*}{d\sigma^2}\right] \geq \sum_{t=1}^{n} \sum_{e \in \mathcal{E}_t^o} \log\left(1 + \frac{z_{t,e}^2}{\sigma^2}\right). \tag{44}$$

Notice that $z_{t,e}^2 = x_e^\mathsf{T} \mathbf{M}_{t-1}^{-1} x_e \leq x_e^\mathsf{T} \mathbf{M}_0^{-1} x_e = \|x_e\|_2^2 \leq 1$, thus we have $z_{t,e}^2 \leq \frac{\log\left(1 + \frac{z_{t,e}^2}{\sigma^2}\right)}{\log\left(1 + \frac{1}{\sigma^2}\right)}$. [5] Hence we have

$$\sum_{t=1}^{n} \sum_{e \in \mathcal{E}_t^o} z_{t,e}^2 \leq \frac{1}{\log\left(1 + \frac{1}{\sigma^2}\right)} \sum_{t=1}^{n} \sum_{e \in \mathcal{E}_t^o} \log\left(1 + \frac{z_{t,e}^2}{\sigma^2}\right) \leq \frac{dE_* \log\left[1 + \frac{nE_*}{d\sigma^2}\right]}{\log\left(1 + \frac{1}{\sigma^2}\right)}. \tag{45}$$

**Remark 2** *When the feature matrix $\mathbf{X} = \mathbf{I}$, we have $d = |\mathcal{E}|$,*

$$\det[\mathbf{M}_n] = \prod_{t=1}^{n} \prod_{e \in \mathcal{E}_t^o} \left(1 + \frac{z_{t,e}^2}{\sigma^2}\right), \quad \text{and} \quad |\mathcal{E}| \log\left[1 + \frac{nE_*}{|\mathcal{E}|\sigma^2}\right] \geq \sum_{t=1}^{n} \sum_{e \in \mathcal{E}_t^o} \log\left(1 + \frac{z_{t,e}^2}{\sigma^2}\right).$$

*This implies that*

$$\sum_{t=1}^{n} \sum_{e \in \mathcal{E}_t^o} z_{t,e}^2 \leq \frac{|\mathcal{E}| \log\left[1 + \frac{n}{\sigma^2}\right]}{\log\left(1 + \frac{1}{\sigma^2}\right)}, \tag{46}$$

*since* $E_* \leq |\mathcal{E}|$.

Finally, from Cauchy-Schwarz inequality, we have that

$$
\sum_{t=1}^{n} \sum_{e \in \mathcal{E}} \mathbf{1}\{O_t(e)\} N_{\mathcal{S}_t,e} \sqrt{x_e^\intercal \mathbf{M}_{t-1}^{-1} x_e} = \sum_{t=1}^{n} \sum_{e \in \mathcal{E}_t^o} N_{\mathcal{S}_t,e} z_{t,e}
$$

$$
\leq \sqrt{\left(\sum_{t=1}^{n} \sum_{e \in \mathcal{E}_t^o} N_{\mathcal{S}_t,e}^2\right)\left(\sum_{t=1}^{n} \sum_{e \in \mathcal{E}_t^o} z_{t,e}^2\right)}
$$

$$
= \sqrt{\left(\sum_{t=1}^{n} \sum_{e \in \mathcal{E}} \mathbf{1}\{O_t(e)\} N_{\mathcal{S}_t,e}^2\right)\left(\sum_{t=1}^{n} \sum_{e \in \mathcal{E}_t^o} z_{t,e}^2\right)}. \quad (47)
$$

Combining this inequality with the above bounds on $\sum_{t=1}^{n} \sum_{e \in \mathcal{E}_t^o} z_{t,e}^2$ (see Equations 45 and 46), we obtain the statement of the lemma. $\qquad\square$

### A.4 Proof of Lemma 2

**Proof:** We use $\mathcal{E}_t^o$ denote the set of edges observed at round $t$. The first observation is that we can order edges in $\mathcal{E}_t^o$ based on breadth-first search (BFS) from the source nodes $\mathcal{S}_t$, as described in Algorithm 2, where $\pi_t(\mathcal{S}_t)$ is an arbitrary conditionally deterministic order of $\mathcal{S}_t$. We say a node $u \in \mathcal{V}$ is a *downstream neighbor* of node $v \in \mathcal{V}$ if there is a directed edge $(v,u)$. We also assume that there is a fixed order of downstream neighbors for any node $v \in \mathcal{V}$.

---

**Algorithm 2** Breadth-First Sort of Observed Edges

**Input:** graph $\mathcal{G}$, $\pi_t(\mathcal{S}_t)$, and $\mathbf{w}_t$

**Initialization:** node queue queueN $\leftarrow \pi_t(\mathcal{S}_t)$, edge queue queueE $\leftarrow \emptyset$, dictionary of influenced nodes dictN $\leftarrow \mathcal{S}_t$

**while** queueN is not empty **do**
    node $v \leftarrow$ queueN.dequeue()
    **for** all downstream neighbor $u$ of $v$ **do**
        queueE.enqueue($(v,u)$)
        **if** $\mathbf{w}_t(v,u) == 1$ and $u \notin$ dictN **then**
            queueN.enqueue($u$) and dictN $\leftarrow$ dictN $\cup \{u\}$
**Output:** edge queue queueE

---

Let $J_t = |\mathcal{E}_t^o|$. Based on Algorithm 2, we order the observed edges in $\mathcal{E}_t^o$ as $a_1^t, a_2^t, \ldots, a_{J_t}^t$. We start by defining some useful notation. For any $t = 1, 2, \ldots$, any $j = 1, 2, \ldots, J_t$, we define

$$
\eta_{t,j} = \mathbf{w}_t(a_j^t) - \overline{w}(a_j^t).
$$

One key observation is that $\eta_{t,j}$'s form a martingale difference sequence (MDS).[6] Moreover, $\eta_{t,j}$'s are bounded in $[-1, 1]$ and hence they are conditionally sub-Gaussian with constant $R = 1$. We further define that

$$
\mathbf{V}_t = \sigma^2 \mathbf{M}_t = \sigma^2 \mathbf{I} + \sum_{\tau=1}^{t} \sum_{j=1}^{J_\tau} x_{a_j^\tau} x_{a_j^\tau}^\intercal, \text{ and}
$$

$$
Y_t = \sum_{\tau=1}^{t} \sum_{j=1}^{J_\tau} x_{a_j^\tau} \eta_{t,j} = B_t - \sum_{\tau=1}^{t} \sum_{j=1}^{J_\tau} x_{a_j^\tau} \overline{w}(a_j^t) = B_t - \left[\sum_{\tau=1}^{t} \sum_{j=1}^{J_\tau} x_{a_j^\tau} x_{a_j^\tau}^\intercal\right] \theta^*.
$$

As we will see later, we define $\mathbf{V}_t$ and $Y_t$ to use the self-normalized bound developed in [1] (see Algorithm 1 of [1]). Notice that

$$\mathbf{M}_t \bar{\theta}_t = \frac{1}{\sigma^2} B_t = \frac{1}{\sigma^2} Y_t + \frac{1}{\sigma^2} \left[ \sum_{\tau=1}^{t} \sum_{j=1}^{J_\tau} x_{a_j^\tau} x_{a_j^\tau}^\mathsf{T} \right] \theta^* = \frac{1}{\sigma^2} Y_t + [\mathbf{M}_t - \mathbf{I}] \theta^*,$$

where the last equality is based on the definition of $\mathbf{M}_t$. Hence we have

$$\bar{\theta}_t - \theta^* = \mathbf{M}_t^{-1} \left[ \frac{1}{\sigma^2} Y_t - \theta^* \right].$$

Thus, for any $e \in \mathcal{E}$, we have

$$\left| \langle x_e, \bar{\theta}_t - \theta^* \rangle \right| = \left| x_e^\mathsf{T} \mathbf{M}_t^{-1} \left[ \frac{1}{\sigma^2} Y_t - \theta^* \right] \right| \leq \| x_e \|_{\mathbf{M}_t^{-1}} \| \frac{1}{\sigma^2} Y_t - \theta^* \|_{\mathbf{M}_t^{-1}}$$

$$\leq \| x_e \|_{\mathbf{M}_t^{-1}} \left[ \| \frac{1}{\sigma^2} Y_t \|_{\mathbf{M}_t^{-1}} + \| \theta^* \|_{\mathbf{M}_t^{-1}} \right],$$

where the first inequality follows from the Cauchy-Schwarz inequality and the second inequality follows from the triangle inequality. Notice that $\| \theta^* \|_{\mathbf{M}_t^{-1}} \leq \| \theta^* \|_{\mathbf{M}_0^{-1}} = \| \theta^* \|_2$, and $\| \frac{1}{\sigma^2} Y_t \|_{\mathbf{M}_t^{-1}} = \frac{1}{\sigma} \| Y_t \|_{\mathbf{V}_t^{-1}}$ (since $\mathbf{M}_t^{-1} = \sigma^2 \mathbf{V}_t^{-1}$), therefore we have

$$\left| \langle x_e, \bar{\theta}_t - \theta^* \rangle \right| \leq \| x_e \|_{\mathbf{M}_t^{-1}} \left[ \frac{1}{\sigma} \| Y_t \|_{\mathbf{V}_t^{-1}} + \| \theta^* \|_2 \right]. \tag{48}$$

Notice that the above inequality always holds. We now provide a high-probability bound on $\| Y_t \|_{\mathbf{V}_t^{-1}}$ based on self-normalized bound proved in [1]. From Theorem 1 of [1], we know that for any $\delta \in (0, 1)$, with probability at least $1 - \delta$, we have

$$\| Y_t \|_{\mathbf{V}_t^{-1}} \leq \sqrt{2 \log \left( \frac{\det(\mathbf{V}_t)^{1/2} \det(\mathbf{V}_0)^{-1/2}}{\delta} \right)} \quad \forall t = 0, 1, \ldots .$$

Notice that $\det(\mathbf{V}_0) = \det(\sigma^2 \mathbf{I}) = \sigma^{2d}$. Moreover, from the trace-determinant inequality, we have

$$[\det(\mathbf{V}_t)]^{1/d} \leq \frac{\mathrm{trace}(\mathbf{V}_t)}{d} = \sigma^2 + \frac{1}{d} \sum_{\tau=1}^{t} \sum_{j=1}^{J_\tau} \| x_{a_j^\tau} \|_2^2 \leq \sigma^2 + \frac{t E_*}{d} \leq \sigma^2 + \frac{n E_*}{d},$$

where the second inequality follows from the assumption that $\| x_{a_k^t} \|_2 \leq 1$ and the fact $J_t = |\mathcal{E}_t^o| \leq E_*$, and the last inequality follows from $t \leq n$. Thus, with probability at least $1 - \delta$, we have

$$\| Y_t \|_{\mathbf{V}_t^{-1}} \leq \sqrt{d \log \left( 1 + \frac{n E_*}{d \sigma^2} \right) + 2 \log \left( \frac{1}{\delta} \right)} \quad \forall t = 0, 1, \ldots, n-1.$$

That is, with probability at least $1 - \delta$, we have

$$\left| \langle x_e, \bar{\theta}_t - \theta^* \rangle \right| \leq \| x_e \|_{\mathbf{M}_t^{-1}} \left[ \frac{1}{\sigma} \sqrt{d \log \left( 1 + \frac{n E_*}{d \sigma^2} \right) + 2 \log \left( \frac{1}{\delta} \right)} + \| \theta^* \|_2 \right]$$

for all $t = 0, 1, \ldots, n-1$ and $\forall e \in E$.

Recall that by the definition of event $\xi_{t-1}$, the above inequality implies that, for any $t = 1, 2, \ldots, n$, if

$$c \geq \frac{1}{\sigma} \sqrt{d \log \left( 1 + \frac{n E_*}{d \sigma^2} \right) + 2 \log \left( \frac{1}{\delta} \right)} + \| \theta^* \|_2,$$

then $P(\xi_{t-1}) \geq 1 - \delta$. That is, $P(\bar{\xi}_{t-1}) \leq \delta$. $\qquad \square$

## Footnotes

[5]Notice that for any $y \in [0,1]$, we have $y \leq \frac{\log\left(1 + \frac{y}{\sigma^2}\right)}{\log\left(1 + \frac{1}{\sigma^2}\right)} \triangleq \kappa(y)$. To see it, notice that $\kappa(y)$ is a strictly concave function, and $\kappa(0) = 0$ and $\kappa(1) = 1$.

[6]Notice that the notion of "time" (or a round) is indexed by the pair $(t,j)$, and follows the lexicographical order. Based on Algorithm 2, at the beginning of round $(t,j)$, $a_j^t$ is conditionally deterministic and the conditional mean of $\mathbf{w}_t(a_j^t)$ is $\overline{w}(a_j^t)$.