[Reviews · NeurIPS 2017]

Reviewer 1



This paper studies an influence maximization problem of a semi-bandit setting in which a player (agent) selects a set of initial active nodes and receives the ids of all the edges that was used to activate one of the end nodes. They consider a model that there is a linear dependence between an edge weight and an edge feature and propose an algorithm called IMLinUCB which uses (alpha,gamma)-approximation oracle as a subroutine. They show an upper bound of (alpha)(gamma)-scaled regret by introducing a new complexity metric called maximum observed relevance that measures a complexity of a given diffusion network in terms of both network topology and edge weights. They also conduct some experiments that support their theoretical result. Nicely written paper. The idea of using the metric called maximum observed relevance to characterize the regret bound is very interesting. Online learning of influence maximization problem itself is interesting but considering a long time of an influence diffusion process, I wonder if it is valuable as a real application.

Reviewer 2



The paper studies the following bandit model: actions are nodes in a known directed graph and at each time step t the following happens: (1) hidden from the player, each edge (i,j) is independently declared "open" or "closed" with fixed but unknown probability w(i,j); (2) the player chooses a subset S_t of nodes of cardinality at most K; (3) all directed paths originated from nodes in S_t that go through open edges are revealed to the player; (4) the player's reward is the number of nodes in these paths. The regret of the player is measured with respect to the expected reward obtained by consistently playing the K-sized subset S of nodes that maximizes the expected reward. Since this set is NP-hard to compute, but easy to approximate, it is assumed that the player has access to an oracle that, given G and estimates of w(i,j) for each edge (i,j), returns the K-sized subset S_t that maximizes the expected reward according to the estimates. The probabilities w(i,j) are assumed to follow a linear model, w(i,j) = x(i,j)*theta, where x(i,j) is the known d-dimensional feature vector associated with edge (i,j) and theta is an unknown parameter vector. A special case is the "tabular model" where all the feature vectors are orthogonal. The paper proposes a natural variant of LinUCB to learn the vector theta using the oracle to compute the maximizing subset. The regret scales with dC_*\sqrt{|E|T} where d is the dimension of theta, E is the edge set, and C_* is a parameter that depends on the interaction between G and w. The experimental section is properly done. The paper builds on a series of works that study influence maximization in the bandit model. The main new ideas are the linear model for probabilities, which provides a key to scalability, and the introduction of C_* as a natural scaling parameter for the regret. The results are not extremely strong, but they are nevertheless interesting and technically solid. The parameter sigma plays no role and could perhaps be omitted from the analysis.